# Molecular mechanisms underlying the BIRC6-mediated regulation of apoptosis and autophagy

Shuo-Shuo Liu[1,5], Tian-Xia Jiang [1,5], Fan Bu[1,5], Ji-Lan Zhao[1,5], Guang-Fei Wang[1,5], Guo-Heng Yang[1], Jie-Yan Kong[1], Yun-Fan Qie[1], Pei Wen[2,3], Li-Bin Fan[3], Ning-Ning Li [4] ✉, Ning Gao [4] ✉ & Xiao-Bo Qiu [1,2] ✉

Procaspase 9 is the initiator caspase for apoptosis, but how its levels and activities are maintained remains unclear. The gigantic Inhibitor-of-Apoptosis Protein BIRC6/BRUCE/Apollon inhibits both apoptosis and autophagy by promoting ubiquitylation of proapoptotic factors and the key autophagic protein LC3, respectively. Here we show that BIRC6 forms an anti-parallel U-shaped dimer with multiple previously unannotated domains, including a ubiquitin-like domain, and the proapoptotic factor Smac/DIABLO binds BIRC6 in the central cavity. Notably, Smac outcompetes the effector caspase 3 and the pro-apoptotic protease HtrA2, but not procaspase 9, for binding BIRC6 in cells. BIRC6 also binds LC3 through its LC3-interacting region, probably following dimer disruption of this BIRC6 region. Mutation at LC3 ubiquitylation site promotes autophagy and autophagic degradation of BIRC6. Moreover, induction of autophagy promotes autophagic degradation of BIRC6 and caspase 9, but not of other effector caspases. These results are important to understand how the balance between apoptosis and autophagy is regulated under pathophysiological conditions.

Apoptosis is a type of programmed cell death[1], whereas autophagy primarily supports survival under nutrient-restricted or other stress conditions by removing cargoes including proteins, organelles, and pathogens[2]. The balance between apoptosis and autophagy is critical for normal development, proper tissue function, and disease pathogenesis[3]. Apoptosis is executed by activation of the family of caspases (cysteine aspartic acid-specific proteases), which are first synthesized as weakly active zymogens and then proteolytically cleaved to generate active enzymes during apoptosis[4]. Procaspase 9 is the initiator caspase, which self-cleaves for activation and then activates effector caspases (e.g., caspases 3 and 7), in the intrinsic or mitochondrial apoptotic pathway, where cytochrome c is released from mitochondria to bind Apaf-1 for self-cleavage of procaspase 9 into the active form[1]. The activity of caspases can be inhibited by the Inhibitor-of Apoptosis Proteins (IAPs), which contain one to three tandem baculoviral IAP repeats (BIRs)[5,6]. The actions of IAP proteins are counteracted by certain pro-apoptotic factors, especially the mitochondria-derived pro-apoptotic factor Smac/DIABLO. Cytosolic Smac precursor is proteolytically processed in mitochondria into its mature form, which is released into the cytosol to interact with IAPs with its exposed IAP-binding motif (IBM) in response to apoptotic stimuli[7,8].

[1]State Key Laboratory of Cognitive Neuroscience & Learning and Ministry of Education Key Laboratory of Cell Proliferation & Regulation Biology, College of Life Sciences, Beijing Normal University, 19 Xinjiekouwai Avenue, Beijing 100875, China. [2]State Key Laboratory of Natural Medicines, China Pharmaceutical University, Nanjing, Jiangsu 211198, China. [3]College of Life Sciences, Anhui Medical University, Hefei, Anhui 230032, China. [4]State Key Laboratory of Membrane Biology, Peking-Tsinghua Joint Center for Life Sciences, School of Life Sciences, Peking University, Beijing 100871, China. [5]These authors contributed equally: Shuo-Shuo Liu, Tian-Xia Jiang, Fan Bu, Ji-Lan Zhao, Guang-Fei Wang. ✉e-mail: ningningli@pku.edu.cn; gaon@pku.edu.cn; xqiu@bnu.edu.cn

The only essential IAP is the exceptionally large (~530 kDa) membrane-associated protein BIRC6 (also referred to as BRUCE or Apollon), which has a ubiquitin-conjugating (UBC) domain and a single BIR[9]. BIRC6 promotes ubiquitylation and degradation of the mature Smac and effector caspases by serving as both ubiquitin-conjugating enzyme (E2) and ubiquitin ligase (E3)[10,11]. We demonstrated previously that BIRC6, unlike other IAPs, also binds to the precursor of Smac and promotes its degradation in an IBM-independent manner[12]. During macroautophagy (referred to as autophagy), cargoes are sequestered into a double-membrane autophagosome[13], which is then fused with the endosome or lysosome to form an autolysosome for cargo degradation[14,15]. LC3-I on the phagophore membrane is conjugated to phosphatidylethanolamine to form LC3-II, which is required for the formation of autophagosomes and selective recruitment of cargoes. We showed previously that BIRC6 together with the proteasome activator PA28γ, which usually promotes the ubiquitin-independent protein degradation[16], promotes proteasomal degradation of LC3-I and thus inhibits autophagy[17]. Interestingly, BIRC6 together with the ubiquitin-activating enzyme (E1) UBA6 promotes ubiquitylation of LC3 at K51 residue[18], though the role of this ubiquitylation is unclear.

Although BIRC6 is a gigantic 530 kDa protein, only two small domains, i.e., the 8 kDa BIR domain and the 19 kDa UBC domain, were previously annotated (Fig. 1a)[10]. It is unknown how BIRC6 interacts with such a variety of binding proteins and how BIRC6 utilizes its ubiquitylation function to regulate both apoptosis and autophagy. Here, we reveal multiple domains in the 3.6-Å Cryo-EM structure of BIRC6, which forms an anti-parallel U-shaped dimer, in addition to the two sets of BIR and UBC located next to each other, and provide the molecular mechanisms for the BIRC6-mediated regulation of apoptosis and autophagy.

## Results

### BIRC6 forms dimer with multiple domains

To characterize the structure of BIRC6 and to understand the structure-function relationship, we expressed and purified the full-length mouse BIRC6 using HEK293F cells, and determined the Cryo-EM structure of BIRC6 at a global resolution of 3.6 Å (Supplementary Figs. 1 and 2, and Supplementary Table 1). The core regions were well resolved at side-chain resolution, allowing an accurate sequence assignment during modeling (Fig. 1b–d, and Supplementary Fig. 2e, f).

The overall structure of BIRC6 is organized as a symmetric homodimer in a U-shape (Fig. 1b–g). The two monomers interact with each other side-by-side in an anti-parallel fashion (Fig. 1b–g), forming a very extensive dimer interface. The central section of BIRC6 is responsible for dimer interactions mediated by an armadillo-repeat domain (ArmRD)[19], which is composed of 31 armadillo repeats (ArmR) (Fig. 1a–g and Supplementary Fig. 3a). Unlike typical armadillo repeats, ArmRD of BIRC6 contains many insertion sequences located either in an inter-ArmR or intra-ArmR manner. Most of these insertions are flexible and not fully resolved in the map. Several insertion domains or motifs, due to their participation in the inter- or intra-molecular interaction, are stabilized and could be modeled unambiguously (Fig. 1a–g and Supplementary Fig. 3d–m).

Within ArmR1, an insertion forms a β-sandwich motif with its core folded as a "beta-jelly-roll" topology[20] (Fig. 1a, h and Supplementary Fig. 3d, e). Since no significant structural homology was identified using the DALI server[21], this insertion was termed jelly-roll-like domain 1 (JRL1) hereafter. Similarly, another jelly-roll-like domain (JRL2) is inserted in ArmR5 (Fig. 1a, h and Supplementary Fig. 3f, g), intermediately followed by a coiled-coil motif (CC) (Fig. 1a, d, g, i and Supplementary Fig. 3h). Next, a long helical motif (LH) inserted between ArmR11 and ArmR12 interacts tightly with ArmR2-7, JRL1 and JRL2 of the same monomer, contributing to the stabilization of JRL1 and JRL2 (Fig. 1a, h, i and Supplementary Fig. 3i). Further towards the C-terminus, a third jelly-roll like the domain is inserted in ArmR18 (Fig. 1a,

j). DALI search indicates that it is a domain of carbohydrate-binding module family 32 (CBM32) (Supplementary Fig. 3j, k)[22–24]. The CBM32 has no contact with the ArmRs from the same BIRC6 monomer, except that a short loop (SL) from ArmR15 stretches to form an additional β-strand to complement one β-sheet of CBM32 (Fig. 1j).

At the C-terminal region of ArmRD, a ubiquitin-like domain (UBL)[25–27] inserted in ArmR26 makes extensive interactions with ArmR20-21 and ArmR24-25 (Fig. 1a, b, e and Supplementary Fig. 3a, l, m). Consequently, these interactions with the UBL generate a sharp bending around ArmR21-24, leading to the formation of the U-shape of the global structure (Supplementary Fig. 3a). In addition, the E1 UBA6 was highly enriched in the BIRC6 complex (Supplementary Fig. 1e and Supplementary Data 1). Unlike UBA1, which binds the C-terminal gly-cine of ubiquitin only through the thioester bond, UBA6 probably binds ubiquitin not only through the thioester bond, but also through non-thioester bonds, though the mechanism remains unclear[28,29]. Notably, the transfected UBL domain of BIRC6 could co-immunoprecipitate with UBA6 from cell lysates (Supplementary Fig. 3b). Thus, we speculate that UBA6 possibly binds BIRC6 through the UBL domain.

The major component of the N-terminal region is a WD40 repeat (WDR)[30] β-propeller domain (Fig. 1a, h and Supplementary Fig. 3c). There is a long insertion between the second and the third blades of the WDR, which folds into two structural domains, BIR and a tightly associated domain (Fig. 1a and Supplementary Fig. 3c). This domain mediates the indirect interaction between BIR and WDR and thus termed BIR-stabilizing domain (BSD) hereafter (Fig. 1a and Supplementary Fig. 3c). All the three domains, WDR, BSD and BIR, interact with the JRL1 domain, orienting the BIR domain toward the center of the U-shape structure (Fig. 1h). The N-terminal section of BIRC6 is dynamic relative to the central section (Supplementary Fig. 2g), which might be beneficial for its substrate capturing. The C-terminal UBC domain (Fig. 1a) is highly flexible and completely invisible in the density map. Considering that the C-terminal end of ArmRD is positioned close to the central space of the U-shaped structure, the UBC domain could be close to the BIR domain (Fig. 1b, e and Supplementary Fig. 3a).

The dimerization of BIRC6 is mediated by the two ArmRDs exclusively. There are extensive side-by-side interactions between aligned ArmRs from the two monomers. In addition, two ArmR insertions also participate in the dimerization (Fig. 1c, f). One is CBM32 inserted in ArmR18, which forms extensive interactions with the ArmR7-13 and CBM32 of the other monomer (Fig. 1j). The other is the CC inserted in ArmR5, which is sandwiched between the region of ArmR1-6 of one monomer and ArmR22-27 of the other to bridge the interaction between the N-terminal and C-terminal segments of two BIRC6 monomers (Fig. 1d, g, i).

In summary, the Cryo-EM structure reveals the domain organization of BIRC6 and a central space for potential substrate accommodation. This anti-parallel arrangement of two monomers would bring close the two sets of functional domains (such as BIR and UBC), suggesting that BIRC6 dimer could work either in cis or in trans to fulfill its role in substrate ubiquitylation.

### Mechanisms by which Smac binds BIRC6

During image processing, we noticed some residual unassigned densities above the two CBM32 domains (Supplementary Fig. 2c, d). Considering that both BIR and UBC domains are facing the central cavity, the residual density might be from endogenous substrates of BIRC6. With several rounds of local 3D classification on this region, several classes with strong additional density above two CBM32 domains were obtained. Refinement of one class led to a 4.7-Å density map in which the extra density was resolved partially at the secondary structural level. Mass-spectrometric analysis of our BIRC6 samples indicated the presence of Smac as a co-purified substrate (Supplementary Fig. 1e and Supplementary Data 1). Furthermore, Myc-Smac

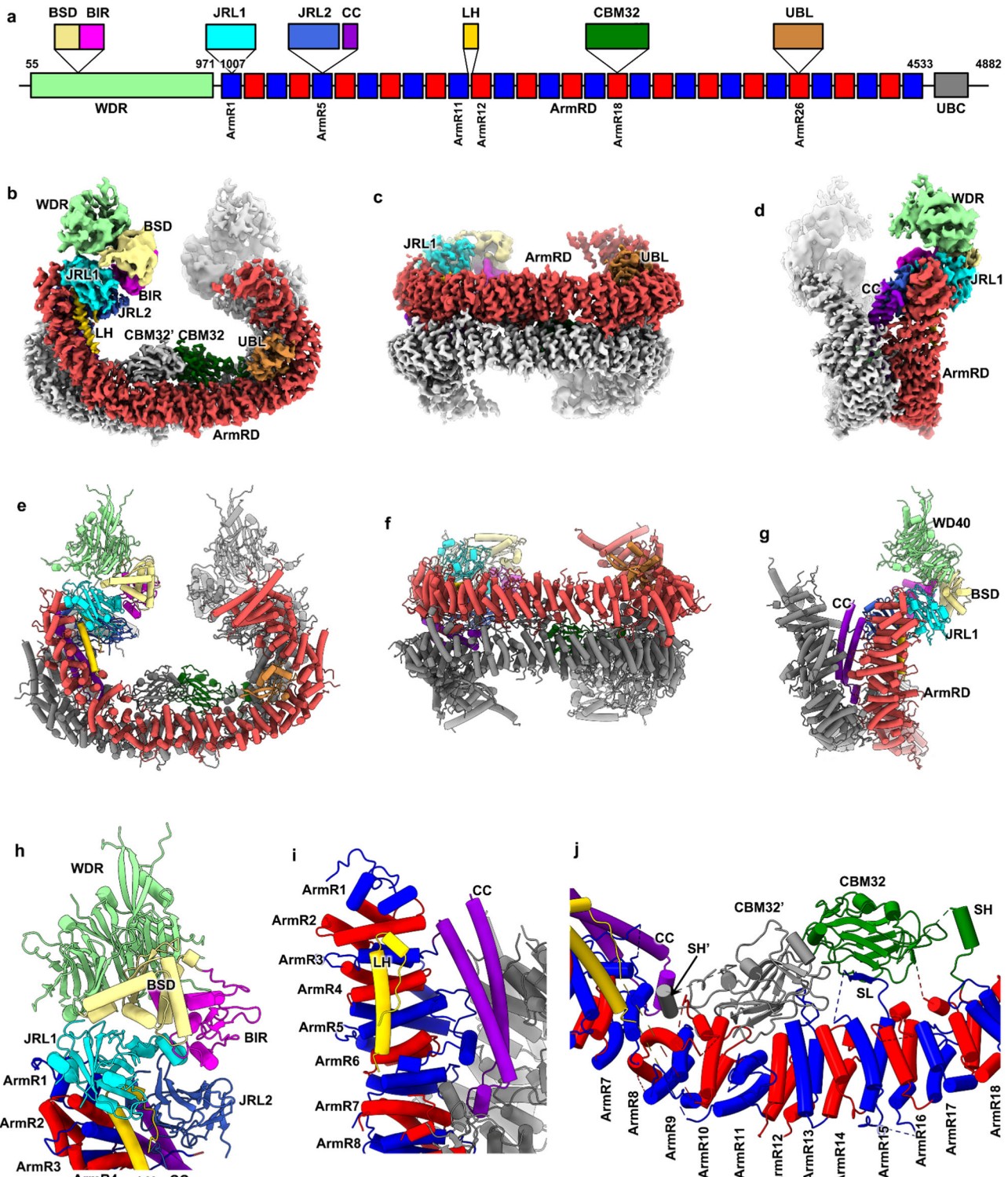

**Fig. 1 | BIRC6 forms an anti-parallel U-shaped dimer with multiple protein-binding domains. a** Schematic domain organization of mouse BIRC6. The 31 ArmRs are colored blue and red alternatively. WDR, WD40 repeat; BSD, BIR-stabilizing domain; BIR, Baculovirus IAP Repeat; JRL1, jelly-roll like domain 1; JRL2, jelly-roll like domain 2; ArmRD, armadillo-repeat domain; CC, coiled-coil motif; LH, long helical motif; CBM32, domain of carbohydrate-binding module family 32; UBL, ubiquitin-like domain; UBC, ubiquitin-conjugating domain. **b**–**d** Composite Cryo-EM density map of BIRC6 dimer displayed in three different views. For one monomer, ArmRD are colored valentine red, and the other domains are color-coded as that in **a**. The other monomer is colored gray. **e**–**g** Same as (**b**–**d**) but showing the atomic model in cartoon representation. **h** Zoomed-in view of the boxed region in (**e**) but with a slight rotation. **i** Zoomed-in view of the boxed region in **g** but with a 180°-rotation. **j** Zoomed-in view of the boxed region in (**e**) but with a slight rotation. The domains in **h**–**j** are colored-coded as that in **a**.

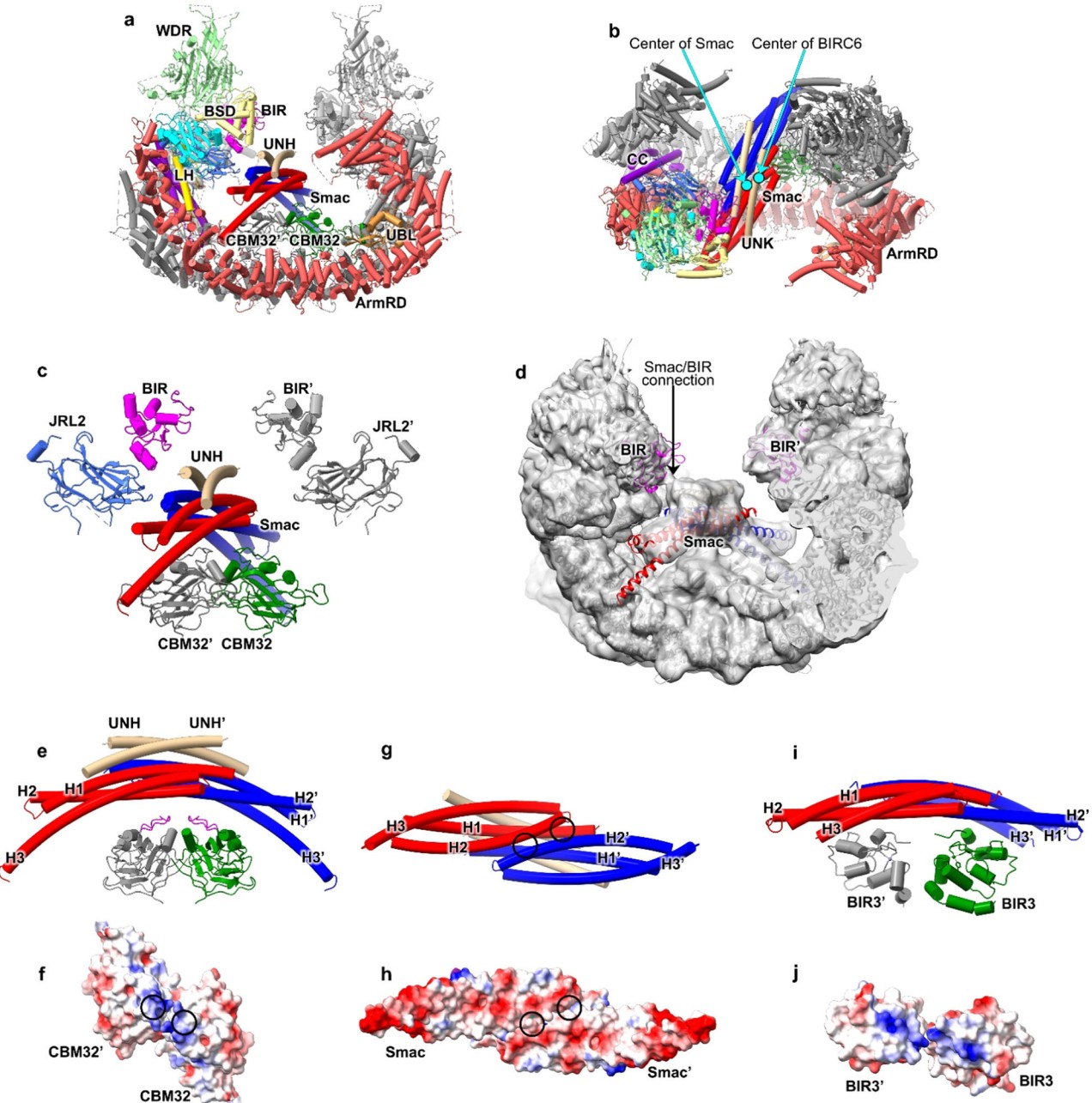

**Fig. 2 | The structure of Smac-BIRC6 complex. a**, **b** Global structure of Smac-BIRC6 complex displayed in two different views in cartoon representation. The domains of BIRC6 are color-coded as that in (Fig. 1b–g). The two monomers of Smac are colored red and blue, respectively. The two UNHs (unknown helix) are colored brown. The two cyan dots in **b** indicate the centers of Smac dimer and BIRC6 dimer, respectively. **c** Structure of Smac-BIRC6 complex showing only Smac and adjacent domains BIRs, JRL2s and CBM32s of BIRC6 dimers. The domains or subunits are color-coded as that in (**a**, **b**). **d** Cryo-EM density map of Smac-BIRC6 complex displayed in low counter level, and with atomic model superimposed. The two Smac monomers are colored red and blue, respectively. The two BIRs are colored magenta. The density connecting Smac and BIR is indicated. **e** Structure of Smac interacting with CBM32, shown in cartoon representation. The domains or subunits are color-coded as that in (**a**, **b**). The loops in the two CBM32s interacting with Smac are colored magenta. UNHs and the helices of Smac are labeled as indicated. **f** The electrostatic potential surface of the CBM32 dimer. The two sites contacting Smac are indicated by black circles. **g** Structure of Smac dimer shown in cartoon representation, in a view with a 90°-rotation to that in panel. The sites contacting CBM32s are indicated by black circles. **h** Same as (**g**) but showing the electrostatic potential surface of the Smac dimer. **i** Structure of the dimeric Smac-XIAP-BIR3 complex by aligning the Smac in monomeric Smac-XIAP-BIR3 complex to Smac in this study. **j** The electrostatic potential surface of the two XIAP-BIR3s, displayed in a view with a 90°-rotation to that in **i** to show the electrostatic potential of the interface to Smac.

could be co-eluted with FLAG-BIRC6 in the gel filtration assay by immunoblotting following SDS-PAGE (Supplementary Fig. 1f). Accordingly, a rigid-body fitting of the dimeric Smac complex (Fig. 2d)[31] reports a good match between the model and the density. In addition to the density of the Smac dimer, two additional single α-helices (termed unknown helix, UNH) with an apparent C2 symmetry could be found to stack on the Smac dimer (Fig. 2a–e). The two helices might each originate from one BIRC6 monomer, and given their binding position, they might be important to promote the binding of Smac.

We noticed that the Smac dimer is slightly off-axial with the BIRC6 dimer (Fig. 2a–c). Smac dimer leans to one side of the central cavity and close to the BIR and JRL2 domains of this side (Fig. 2a–c). Smac usually binds to the BIR domain of other IAPs with the N-terminal IBM to antagonize the inhibition of IAP to caspases[32]. Although the interaction of Smac-IBM with BIR could not be identified due to the resolution limitation, the density connection between Smac and one BIR domain could be observed when the density map was shown in lower contour level (Fig. 2d). In contrast, the Smac connection to the other BIR domain was much weaker (Fig. 2d), indicating a potentially complex regulation of Smac on the two BIR domains.

The stabilization of Smac is mainly dependent on the interactions with the CBM32 dimer, which is mediated by the helices H2/H2′ of Smac dimer and the short loops (residues 3172–3182) of the CBM32 dimer in electrostatic interactions (Fig. 2a, c, e–h). The regions of Smac contacting the two CBM32 domains are slightly different, due to the symmetry mismatch between BIRC6 and Smac (Fig. 2g, h). In addition to the principal interaction of Smac-IBM with the BIR domain, another interface between Smac and BIR3 of XIAP was previously also reported[32]. By structural superimposition, it is surprising to find that this second interface is similar to that of Smac-CBM32 interaction and both are mediated by electrostatic interactions (Fig. 2i, j). CBM32 might act as a functional equivalent of BIR3 of XIAP to build the second Smac interface, which could explain why BIRC6 has only one BIR domain while XIAP has three[33].

This binding pattern of Smac shows that the central space of the BIRC6 dimer is the substrate binding pocket. Indeed, some other unassigned residual densities were also observed in the central space of the non-Smac-binding groups[34–36] (Supplementary Fig. 2i). Given that various domains and insertions are facing this cavity, BIRC6 ought to have a complex regulation on the recognition of its substrates.

## Smac cannot reduce procaspase 9 binding

We showed previously that BIRC6 directly inhibits the activity of the purified caspase 9, but not the purified effector caspase 3[12]. Although BIRC6 binds the purified processed caspase 9[12], it remains unclear whether it associates with procaspase 9 in cells. Since the transfected wild-type procaspase 9 was quickly processed (Fig. 3a), we constructed the non-cleavable procaspase 9 triple mutant (E306A−D315A−D330A; Procasp9-3M) and the quadruple mutant with additional active site mutation (E306A−D315A−D330A−C287A; Procasp9-4M). Transfected Myc-BIRC6 could co-immunoprecipitate with both triple and quadruple mutants of FLAG-procaspase 9 (Fig. 3a). Accordingly, the purified BIRC6 could partially co-elute with His-procasp9-4M in the gel filtration assay (Supplementary Fig. 4a, b). Overexpression of the Myc-tagged wild-type Smac did not affect the association of BIRC6 with the FLAG-tagged quadruple mutants of procaspase 9, but markedly reduced their association with the pro-apoptotic protease HtrA2/Omi or procaspase 3 (Fig. 3b–e), hinting that BIRC6 binds procaspase 9 in a mechanism different from that for HtrA2 or procaspase 3. All IBM-containing proteins must be processed to expose their N-terminal IBM for binding regular IAPs, such as XIAP[34]. However, BIRC6, unlike other IAPs, also binds to the Smac precursor with unexposed IBM and promotes its degradation in an IBM-independent manner[12]. Although caspase 9, Smac, and HtrA2 all share similar IBMs, whether HtrA2 binds BIRC6 in an IBM-independent manner was still unclear. Smac with IBM mutations, where the AVPI stretch was mutated into AAAI, could not reduce the co-immunoprecipitation of BIRC6 with HtrA2 (Fig. 3b, d), suggesting that the IBM of HtrA2 should also mediate its association with BIRC6. In contrast, this Smac mutant did not affect the co-immunoprecipitation of BIRC6 with procaspase 9, though reducing the co-immunoprecipitation of XIAP with procaspase 3 (Fig. 3b, c, e). On the other hand, the association of active caspase 9 with BIRC6 could be outcompeted by the wild-type Smac, instead of its AVPI mutant, in a fashion similar to that of active caspase 3 (Fig. 3f). We then

showed that the purified BIRC6 could strongly inhibit the activity of caspase 9, but only weakly for caspase 3, in the HEK293T cell lysates following caspase activation by cytochrome c and dATP (Fig. 3h and Supplementary Fig. 4c–e). Thus, Smac outcompetes caspase 3 and HtrA2, but not procaspase 9, for binding BIRC6 in cells, while BIRC6 strongly inhibits caspase 9 and weakly inhibits caspase 3.

## Caspase 9 can be degraded by autophagy

Nutrient starvation and rapamycin inhibit the activity of mTOR, and induce autophagy[37]. Starvation in Hanks' Balanced Salt Solution (HBSS) or treatment with rapamycin reduced the levels of procaspase 9, but not of procaspase 3 and procaspase 7, in MDA-MB-453 cells (Fig. 4a, b, Supplementary Fig. 5a, b). The levels of procaspase 9 decreased gradually upon starvation in COS7 cells, especially at 4 h, and were reversed by the autophagy inhibitor Bafilomycin A1 (Baf A1) (Fig. 4c, d). Moreover, the levels of active caspase 9, though very low, also decreased upon starvation and could be reversed by Baf A1 (Fig. 4c, d). In the HEK293T cells transfected with Myc-procasp9-3M, the half-life of the Myc-procasp9-3M was about 12 h, but was reduced to less than 8 h in the presence of rapamycin (Supplementary Fig. 5c, d). Atg5 is essential for the formation of autophagosome[38]. The treatment with rapamycin reduced the levels of procaspase 9 in the wild-type mouse embryonic fibroblast (MEF) cells, but not in the Atg5-deficient cells (Fig. 4e, f). At the same time, the levels of BIRC6, which usually degrades during autophagy[17], could be reduced by the treatments of rapamycin or serum starvation in MDA-MB-453 cells (Fig. 4g, h). Further, little caspase 9 co-localized with LC3-II under normal condition, but its co-localization with LC3-II increased dramatically upon starvation (Fig. 4i). These results suggest that both procaspase 9 and active caspase 9, together with BIRC6, are degraded by autophagy upon autophagic stimulation.

## BIRC6 binds LC3 through LIR

Because BIRC6 associates with both caspase 9 and LC3, the autophagic degradation of caspase 9 could potentially be mediated by BIRC6. Thus, we attempted to explore the mechanisms by which BIRC6 interacts with LC3. Proteins usually bind LC3 through a LC3-interacting region (LIR) with the consensus sequence, X3X2X1[W/F/Y]X1X2[L/I/V]X4X5, where alternative letters are placed in square brackets[39]. We showed previously that BIRC6 contains a putative LIR (amino acids 4670-NPQ TSS FLQV LV-4681 in mice) and co-immunoprecipitates with LC3[17]. Unexpectedly, addition of this putative LIR peptide (i.e., LIR1, NPQTSSFLQVLV) at 100 μM could not reduce the precipitation of LC3 with BIRC6 (Fig. 5a). Conversely, the LIR1 peptide at 100 μM, but not its mutant (NPQTSSALQALV, where the replaced residues are underlined), dramatically increased the amounts of both LC3-I and LC3-II associated with BIRC6 (Fig. 5a). To clarify whether this effect is direct, we purified the precursor of LC3-I (i.e., LC3B), which was expressed in bacteria. When incubated with the purified FLAG-tagged BIRC6, the LIR1 peptide at 100 μM still increased the association of LC3B with BIRC6 in vitro (Fig. 5b). Next, we synthesized a hypothetical universal LIR peptide with a sequence different from any known protein (hu-LIR, EPLDFDWEIVLEEEM), which theoretically competes with any LIR motif[40]. The hu-LIR peptide at 100 μM, but not its mutant (EPLDF-DAEIALEEEM, where the replaced residues are underlined), could reduce the association of LC3 with BIRC6 (Fig. 5c–e). In contrast, the AVPI stretch of Smac or an unrelated peptide could not affect binding of LC3 to BIRC6, though the AVPI stretch could reduce the association of Smac with XIAP (Fig. 5c–e). Because the hu-LIR peptide sequence is generally different from that of the BIRC6 LIR region and cannot bind the corresponding LIR-pairing region of BIRC6, it is thus free to reach the BIRC6 LIR region and block the LC3 binding, which should happen at a low frequency in regularly cultured cells (Open status). We hypothesize that BIRC6-LIR1 peptides competitively and preferentially bind the part of the other BIRC6 molecule (i.e., the LIR-pairing region),

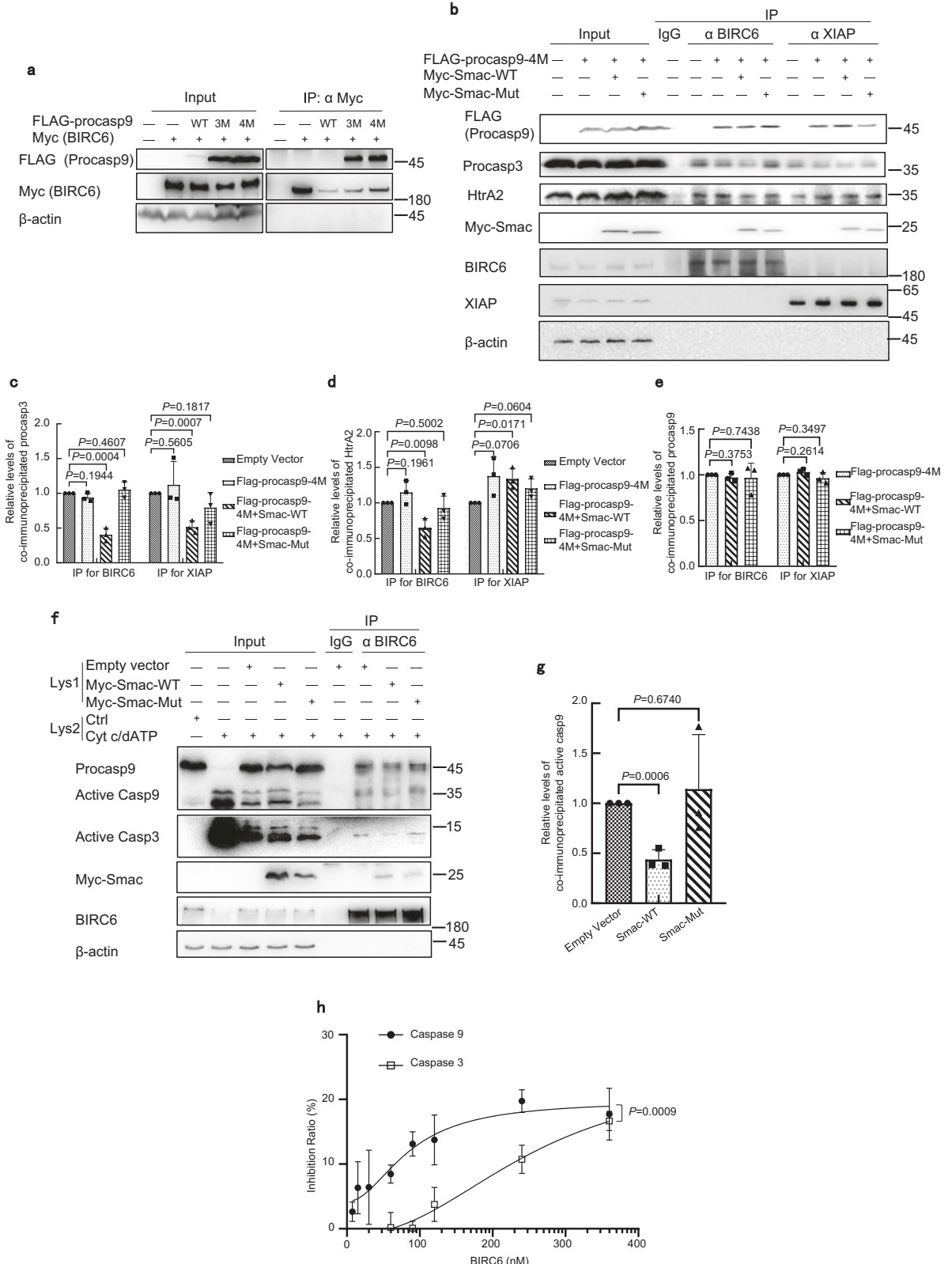

which supposedly binds the LIR region of BIRC6 (Closed status), and then make the BIRC6 LIR region free to bind LC3. Indeed, the His-tagged N1 region of BIRC6 (aa 27–358) that contains the BIR domain (aa 271–377), but not its N2 region (aa 358–543), could pull-down the LIR1 peptide in vitro, supporting the presence of this LIR-pairing region (Supplementary Fig. 5e). In agreement with the pull-down assays, AlphaFold2 structure prediction revealed an interaction between the

LIR1 peptide and the N1 region of BIRC6, characterized by the formation of an anti-parallel β-sheet interface involving LIR1 and residues VCLV (aa 308–311) of the BIR domain in the N1 region (Supplementary Fig. 5f). Thus, further increase of LIR1 peptide concentrations should competitively bind the relaxed BIRC6 LIR and reduce the binding of LC3. We found that LC3 bound BIRC6 at the highest levels in the presence of 100 µM of LIR1 peptides by using different concentrations of

**Fig. 3 | Smac outcompetes caspase 3 and HtrA2, but not procaspase 9.**
**a** HEK293T cells were co-transfected with Myc-BIRC6 and the FLAG-tagged wild-type (WT), triple mutant (E306A−D315A−D330A; Procasp9-3M) or quadruple mutant procaspase 9 (E306A−D315A−D330A−C287A; Procasp9-4M). Immunoblotting was performed following immunoprecipitation with anti-Myc antibodies, and was repeated twice independently with similar results. **b−e** HEK293T cells were transfected with the FLAG-tagged Procasp9-4M in combination with Myc-tagged wild-type or mutant Smac. Immunoblotting was performed following immunoprecipitation with anti-BIRC6 or XIAP antibodies in the presence of 50 μM of the pan-caspase inhibitor Z-VAD-fmk (**b**). Relative levels of co-immunoprecipitated procaspase 3, procaspase 9, and HtrA2 were quantified by densitometry and normalized to procaspase 3 (**c**), HtrA2 (**d**), and procaspase 9 (**e**) and in the input with three independent biological replicates. **f, g** The lysates of HEK293T cells transfected with the Myc-tagged wild-type or mutant Smac (Lys1) were mixed with HEK293T cell extracts (Lys2), where caspases were activated with Cyt c and dATP, in the presence of 50 μM of the pan-caspase inhibitor Z-VAD-fmk. Immunoblotting

was performed following immunoprecipitation with anti-BIRC6 antibodies (**f**). Overexpression of the proapoptotic Smac reduced the cellular levels of procaspase 9, but increased those for active caspase 9, leading to the reduced levels of pro-caspase 9 co-immunoprecipitated with BIRC6. Relative levels of co-immunoprecipitated active caspase 9 were quantified by densitometry (normalized to active caspase 9 in the input) with three independent biological replicates (**g**). ***$P < 0.001$, one-way ANOVA. ns: not significant. **h** Active caspases in HEK293T cell extracts were incubated with the purified FLAG-tagged BIRC6 at indicated concentrations. The caspase activities were analyzed using the caspase 9 substrate Ac-LEHD-AMC or caspase 3 substrate Z-DEVD-7-AMC. Inhibition ratio of caspase activity by BIRC6 (%) was calculated as described in the Methods. XIAP and β-actin were detected on a different blot with the input loaded for 30 μg/sample from the same experiment. Experiments were performed with three replicates (**b**, **f**, **h**) and are shown as mean ± standard deviation (**c−e**, **g**, **f**). **c−e**, **g** for one-way ANOVA; (**h**) for two-way ANOVA. Source data are provided as a Source Data file.

LIR1 peptides. The binding indeed declined afterwards with the increase of the LIR1 peptide concentrations, reaching levels lower than that in the untreated sample at 400 μM of LIR1 peptides (Fig. 5f, g). Thus, LIR1 peptides increase BIRC6 binding to LC3 at low concentrations, but inhibit this binding at high concentrations. Taken together, we reason that dimerization of BIRC6 usually blocks binding of its LIR motif to LC3, whereas certain cellular signals (e.g., apoptotic or excessive autophagic signals) or the LIR1 peptides dissociate this dimerized region to expose the buried LIR surface for binding LC3, leading to the eventual degradation of LC3 by the proteasomal or autophagic pathway (Fig. 5h).

### K51R mutation of LC3 inhibits apoptosis
We showed previously that monoubiquitylation of the adaptor protein SIP/CacyBP can switch the cell status between apoptosis and autophagy, likely by controlling degradation of BIRC6[17], which serves as both a ubiquitin-conjugating enzyme and ubiquitin ligase (E2/E3) to catalyze monoubiquitylation of LC3 at K51[18]. To test whether the BIRC6-mediated ubiquitylation of LC3 balances apoptosis and autophagy, we constructed an LC3 mutant with the mutation at K51 (K51R), which is the only ubiquitylation site of LC3[18]. This mutation inhibited degradation of the GFP-fused LC3 in HEK293T cells and A549 cells treated with the translation inhibitor cycloheximide (CHX) (Fig. 6a−d). Overexpression of the proteasome activator PA28γ facilitated degradation of the wild-type LC3, but not its K51R mutant, in either HeLa or HEK293T cells (Supplementary Fig. 6a, b). It is noteworthy that the band of the PA28γ was weaker at later time points in LC3-K51R, but not LC3-WT samples. It is well established that proteasomes and their subunits can be degraded by autophagy[41]. These results might suggest that LC3-K51R also promoted autophagic degradation of PA28γ. However, the K51R mutation did not affect the association of LC3 with PA28γ as shown by the co-immunoprecipitation assays (Supplementary Fig. 6c). The K51R mutation markedly increased the ratio of LC3-II/I, a marker for the occurrence of autophagy, in HEK293T cells treated with rapamycin in conjunction with Bafilomycin A1 (Baf A1) to visualize the conversion of LC3-I to LC3-II, because Baf A1 blocks the fusion of autophagosomes with endosomes or lysosomes[14] (Fig. 6e). Accordingly, this mutation promoted autophagic degradation of BIRC6 and the autophagy receptor protein p62 in the HEK293T cells treated with rapamycin (Fig. 6f). The results from the LC3 deletion or knockdown cells further support our notion that the LC3-K51R mutation inhibits the degradation of the transfected GFP-LC3 (Fig. 6g, h and Supplementary Fig. 6d). Moreover, the K51R mutation led to formation of much more autophagosomes, as marked by LC3 puncta, in HeLa cells (Fig. 6i). On the other hand, the LC3 mutation at K51R obviously reduced the levels of the cleaved caspase 3 and apoptosis in response to treatment of the topoisomerase inhibitor etoposide (Fig. 6j, k). These results suggest that K51 of LC3, which is the site for both

acetylation[42] and the BIRC6-mediated ubiquitination[18], is critical to inhibiting autophagy and promoting apoptosis.

## Discussion
An appropriate level of caspase 9 is the key to determining cell death or survival, but how the levels and activities of caspase 9 are maintained remains unclear. The results in this study strongly suggest that the only essential IAP BIRC6 plays a vital role in inhibiting apoptosis by suppressing caspase 9 under normal conditions (Supplementary Fig. 6e). We determine the 3.6-Å Cryo-EM structure of BIRC6, which forms an anti-parallel U-shaped dimer with the two sets of functional domains BIR and UBC located next to each other. Unexpectedly, Smac was highly enriched in the BIRC6 complex in regularly cultured cells (Supplementary Fig. 1e and Supplementary Data 1). A dimer of Smac was found located in the central cavity of a subpopulation of the particles directly purified from cells. BIRC6 binds Smac through one BIR domain, one unknown helix, and a domain of carbohydrate-binding module family 32 (CBM32) in a mechanism different from any other IAPs. Although our data in this study and the results by the other three groups, which appeared when this manuscript was under review[43–45], all demonstrated that BIRC6 forms the strongest complex with Smac among all BIRC6 clients, it is noteworthy that isolations of the stable BIRC6-Smac complex in all these studies were from the cells under normal growth conditions. Either apoptosis or autophagy actually happens under various stress conditions. Procaspase 9 and LC3, both of which are potential targets of BIRC6, are critical to the initiation of apoptosis and autophagy, respectively[4,46–49]. Thus, the BIRC6-mediated regulation of procaspase 9 and LC3 or their regulation by Smac should be important to the balance between apoptosis and autophagy. We showed previously that the adaptor protein SIP/CacyBP might balance apoptosis and autophagy probably by controlling the degradation pathways of BIRC6 under certain stress conditions[17]. This study shows that overexpression of Smac could not exclude procaspase 9 from binding cellular BIRC6, hinting that the primary role of Smac in apoptosis is most likely to release the pro-apoptotic protease HtrA2 and the effector caspases, all of which were enriched in the BIRC6 complex isolated from cells (Supplementary Fig. 1e and Supplementary Data 1). Failure of Smac to release procaspase 9 from BIRC6 presumably allows the sufficient chance for BIRC6 to promote ubiquitylation and proteasomal degradation of procaspase 9 in addition to inhibiting the activity of active caspase 9 directly. This notion is in accord with the fact that the E1 UBA6 was highly enriched in the BIRC6 complex isolated from cells (Supplementary Fig. 1e and Supplementary Data 1).

BIRC6 might target caspase 9 for autophagic degradation in response to autophagic stimuli, because both procaspase 9 and active caspase 9, together with BIRC6, underwent autophagic degradation upon rapamycin treatment or nutrient starvation. Coincidently,

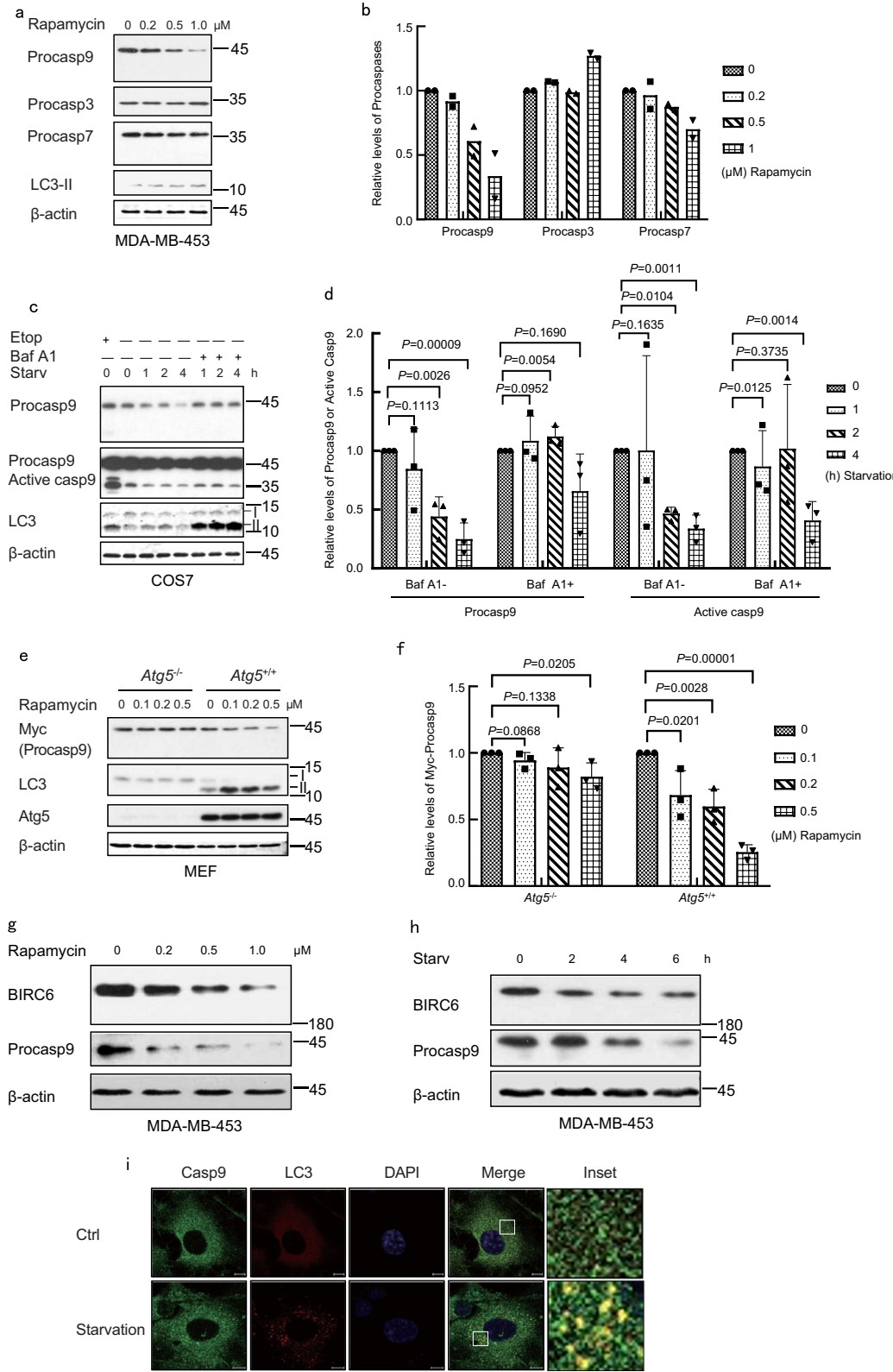

knockdown of the autophagic receptor p62 markedly increases the levels of active caspase 9 and active caspase 3 in the Cadmium-treated JEG-3 trophoblast cells[50]. Unlike autophagic degradation of active caspase 8 or active caspase 3 in response to apoptotic stimuli[51,52], the degradation of the initiator caspase (i.e., caspase 9) during autophagy might hold the effector caspases, such as caspases 3 and 7, in inactive forms, supporting that autophagic degradation of caspase 9 plays an

important role in inhibiting apoptosis under physiological or pathological conditions.

In response to apoptotic stimuli, BIRC6, together with UBA6, should promote ubiquitylation of LC3 at K51, leading to inhibition of autophagy and promotion of apoptosis, because mutation of the LC3 ubiquitylation site promoted autophagy and inhibited apoptosis (Supplementary Fig. 6e). The results from this study further strengthen

**Fig. 4 | Procaspase 9 can be degraded by the autophagy mediated by LC3.**
**a, b** MDA-MB-453 cells were treated with rapamycin at 0, 0.2, 0.5, or 1.0 µM for 24 h.
Protein levels were analyzed by immunoblotting (**a**), and procaspase 9 or procas-
pase 3 or procaspase 7 was quantified by densitometry (normalized to β-actin) with
two independent biological replicates (**b**). **c, d** COS7 cells were starved in HBSS with
or without 10 nM of Baf A1 for 0, 1, 2, or 4 h. Treatment with 20 µM of etoposide for
24 h was served as a control for caspase activation. Protein levels were analyzed by
immunoblotting (**c**), quantified by densitometry (normalized to β-actin) with three
independent biological replicates and are shown as mean ± standard deviation,
one-way ANOVA (**d**). **e, f** *Atg5*[+/+] and *Atg5*[−/−] MEF cells were transfected with Myc-
procaspase 9 and then treated with rapamycin at the indicated concentration (µM)

for 24 h. The protein levels were detected by immunoblotting (**e**), quantified by
densitometry (normalized to β-actin) with three independent biological replicates
and are shown as mean ± standard deviation, one-way ANOVA (**f**). **g** MDA-MB-453
cells were treated with rapamycin at 0, 0.2, 0.5, or 1.0 µM for 24 h. Protein levels
were analyzed by immunoblotting. **h** MDA-MB-453 cells were starved in HBSS for 0,
2, 4, or 6 h. Protein levels were analyzed by immunoblotting. The experiments (**g**, **h**)
were repeated two times with similar results. **i** COS7 cells were starved in HBSS
medium for 0 or 2 h, and then subjected to immunostaining and confocal micro-
scopy. DNAs in nuclei were stained with DAPI. *n* = 30 images, Scale bar: 10 µm. The
white rectangles indicate the enlarged areas. I: LC3-I, II: LC3-II. Source data for (**a**–**h**)
are provided as a Source Data file.

our previous notion that LC3 binds the LIR motif of BIRC6 (4670−NPQ
TSS FLQV LV-4681)[17], which does not overlap with the binding region of
Smac, providing a basis for the coordinated regulation of apoptosis
and autophagy. We speculate that dimerization of BIRC6 blocks the
binding of its LIR motif to LC3, whereas certain cellular signals might
dissociate this dimerized region to expose the originally buried LIR for
binding LC3, leading to eventual degradation of LC3 by the protea-
somal or autophagic pathway. To remove BIRC6, the ubiquitin ligase
Nrdp1 promotes ubiquitylation and proteasomal degradation of BIRC6
in response to apoptotic stimuli, as we showed previously[9]. Accord-
ingly, Nrdp1 was also found in the BIRC6 complex (Supplementary
Fig. 1e and Supplementary Data 1). Given that mutation or over-
expression of BIRC6 is frequently associated with various cancers[53–55],
our findings might contribute to the discovery of therapeutic targets
for the enhancement or inhibition of autophagy and apoptosis.

## Methods

### Purification of BIRC6
The plasmid pCI-Myc-BIRC6 (mouse) was a gift from Dr. Stefan Jentsch
as described[17]. A 3×FLAG tag was then inserted into the plasmid ahead
of the N-terminus of BIRC6 gene. The plasmids were then transiently
transfected into HEK293F expression system for overexpression. After
incubation for 48 h, the cells were collected, washed one with ice-cold
1× PBS, and then resuspended in buffer A (20 mM HEPES-KOH, pH 7.5,
10 mM KCl, 1.5 mM MgCl$_2$, 1 mM EDTA, 1 mM EGTA, 1 mM DTT, 10%
glycerol, 0.1 mM PMSF, 10 mM Disodium β-glycerophosphate penta-
hydrate, 2.5 mM sodium pyrophosphate, 1 mM NaF, 1 mM Na$_3$VO$_4$,
1 µg/mL Benzonase Nuclease and a protease inhibitor mixture). The
resuspended cells were then lysed by sonication, and the lysate
supernatant after centrifugation was incubated with M2-FLAG-affinity
beads (Millipore, #M8823) for 1.5 h at 4 °C. After several round of
washing with buffer B (20 mM Tris-HCl, pH 8.0, 100 mM KCl, 10%
glycerol, 1 mM ZnCl$_2$, 1 mM NaF, 1 mM Na$_3$VO$_4$ and the protease inhi-
bitor mixture), the FLAG-affinity proteins were eluted by 0.5 mg/mL
3×FLAG peptide (MDYKDHDGDYKDHDIDYKDDDDK) dissolved in
buffer C [Phosphate-buffered saline (PBS) supplemented with 100 mM
KCl, pH 7.4]. The elute was further purified by gel filtration (Superose 6
Increase Columns 10/300 GL, Cytiva) using buffer C. The fraction at
0.5 mL/tube was collected starting from 7 mL. To optimize the Cryo-
EM grids preparation for data collection, the peak fractions of BIRC6
were gathered and incubated with 0.01% glutaraldehyde for 10 min,
which was then quenched by 40 mM Tris-HCl, pH 7.4 buffer. The cross-
linked sample was then purified by another round of gel filtration. The
peak fractions were pooled and concentrated for Cryo-EM grid
preparation.

### Purification of His-tagged proteins
1 mM IPTG (Beyotime, #ST098-5g) was used to induce the expression
of His-Caspase 9-4 M (E306A−D315A−D330A−C287A)-pET-30a, His-
LC3B-pET-30a, His-BIRC6-N1 (aa 27−358) and His-BIRC6-N2 (aa
358−543) in BL21 for overnight at 16 °C. After incubation for 16 h, the
cells were collected, washed once with ice-cold PBS. Lysis/binding
/wash buffer contained 300 mM NaCl, 25 mM HEPES, 20 mM

Imidazole, and 1 µg/mL Benzonase at pH 7.5. The resuspended cells
were then lysed by sonication, and the lysate supernatant after cen-
trifugation was incubated with Ni-NTA beads for 20 min at 4 °C. The
His-tagged proteins were eluted by the elution buffer containing
300 mM NaCl, 25 mM HEPES, and 250 mM Imidazole, pH 7.5. The
elute was further purified by gel filtration (Superdex TM 75 10/300 GL Col-
umns, Cytiva) using buffer C (PBS supplemented with 100 mM KCl, pH
7.4). The fraction at 0.5 mL/tube was collected starting from 7 mL.

### In vitro BIRC6 binding assays
LAG-BIRC6 was incubated with about 0.5 mg His-Caspase 9-4M-pET-
30a with the molar ratio at 1:300 for 30 min at RT. About 0.5 mg His-
LC3-pET-30a was incubated with FLAG-BIRC6 in the absence or pre-
sence of 100 µM BIRC6-LIR1 peptides (NPQTSSFLQVLV) for 30 min at
RT. The molar ratio between FLAG-BIRC6 and His-LC3-pET-30a was
1:300. The mixture was then purified by another round of gel filtration
(Superose 6 Increase 10/300 GL Columns, Cytiva).

### Mass spectrometry (MS)
To identify the proteins isolated from cell lysates, the samples were
analyzed by mass spectrometry following the established protocol[56].
Briefly, approximately 50 µg/lane of proteins were loaded onto the
SDS-PAGE gel and stained with Coomassie bright blue. Acetonitrile was
added to dehydrate the gel, which was then incubated with 10 mM
Dithiothreitol/25 mM NH$_4$HCO$_3$ for 1 h at 56 °C, followed by incubation
with 55 mM ICH$_2$CONH$_2$/25 mM NH$_4$HCO$_3$ for 45 min in dark and two
washes with 25 mM NH$_4$HCO$_3$. Acetonitrile was added for dehydrating
again. Then, the gel was crushed into the 25 mM NH$_4$HCO$_3$ buffer
containing trypsin followed by incubation at 37 °C for overnight. 0.1%
formic acid in acetonitrile was added for 10 min, and the supernatant
was transferred to a new tube following a brief centrifugation. The
peptide sample was next dissolved with 0.1% formic acid to reach
0.1 µg/µL. LC-MS/MS detection was performed using EASY-nLC 1200
liquid chromatography column, C18 liquid chromatography column,
and Thermo Q Exactive Plus mass spectrometer. The liquid chroma-
tographic solution A was 0.1% formic acid, and the solution B con-
tained 80% acetonitrile/0.1% formic acid. Mass spectrum MS1 scanning
parameters: (a) Orbitrap resolution was 70000; (b) scanning range was
300−1500 *m/z*; (c) AGC Target was 3,000,000; (d) maximum injection
time was 60 ms. MS2 scanning parameters: (a) Fracture mode was HCD
(30% energy); (b) acquisition mode was Data-dependent; (c) Orbitrap
resolution was 17500; (d) AGC target was 100,000; (e) maximum
injection time was 45 ms; (f) Fixed First Mass was 110 *m/z*; (g) Isolation
Window was 1.6 *m/z*. The results of MS (*n* = 1) were processed by Pro-
teome Discoverer 2.2 software, and aligned with UniProtKB/Swiss-Prot
human proteome database with precursor ion mass tolerance of 10
ppm and fragment ion mass tolerance of 0.02 Da.

### Negative-staining EM
The quality of the purified BIRC6 samples was examined by negative-
staining EM. In brief, 2−4 µL aliquots of samples were applied on the
glow discharged copper grids, and blotted by filter papers after
30−60 s' waiting time. The samples were then washed and stained

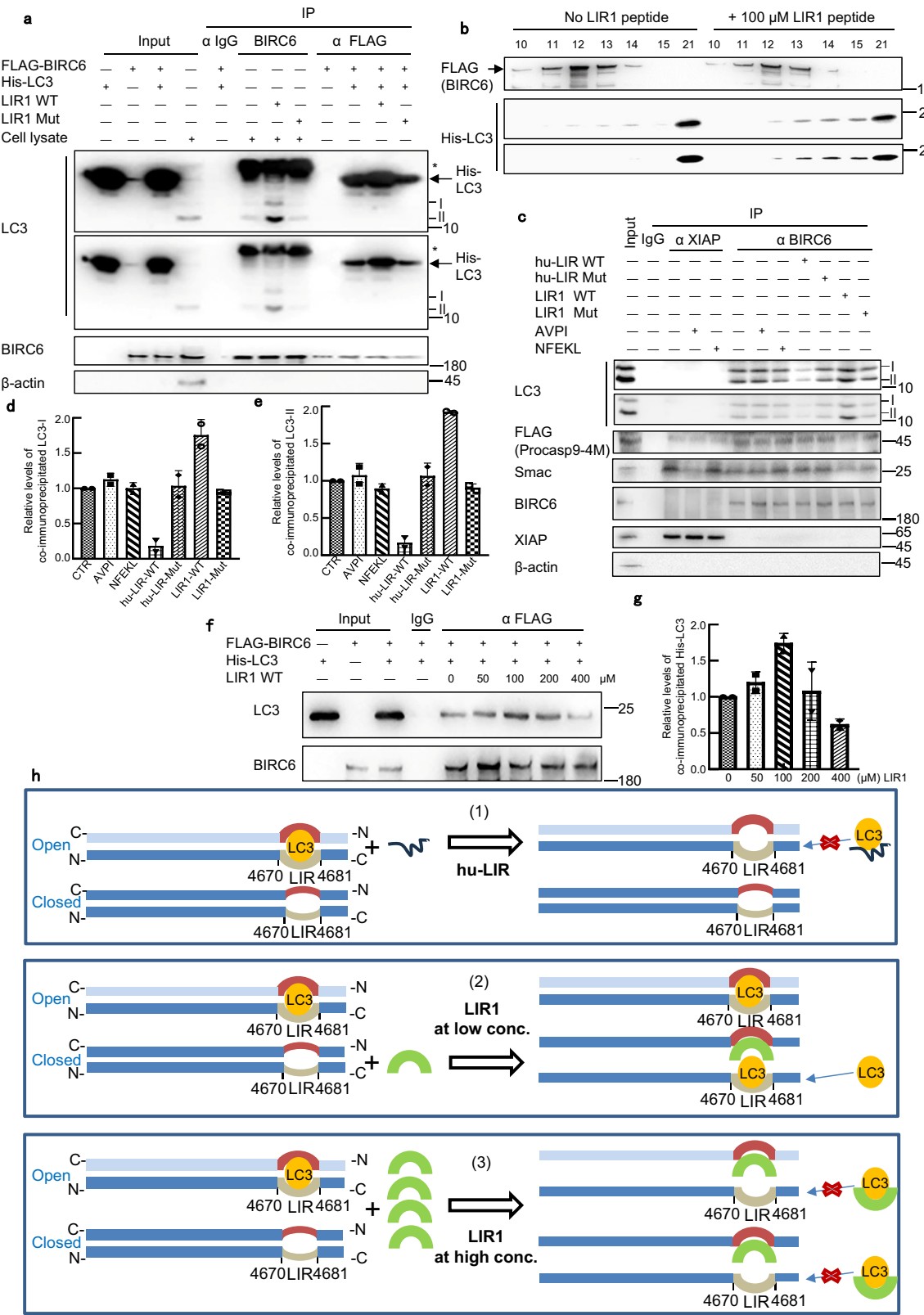

using 2% (w/v) uranyl acetate solution. The grids were screened using a JEOL JEM-1400Flash microscope operated at an accelerating voltage of 80 kV.

**Cryo-EM data collection**

To prepare Cryo-EM grids, 3–4 μL aliquots of samples were applied on the glow-discharged holey carbon gold grids (C-flat, 2/2, 300 mesh),

and flashed frozen in liquid ethane after blotting, using an FEI Vitrobot IV with 100% humidity and 4 °C. The grids were then screened using an FEI Talos Arctica microscope operated at 200 kV. The qualified grids were recovered and transferred into a 300-kV FEI Titan Krios microscope for data collection. The dataset was auto-collected using the SerialEM software[57], at a nominal magnification of ×64,000, corresponding to a calibrated pixel size of 1.37 Å at object scale, with a

**Fig. 5 | LC3 binds the LC3-interacting region of BIRC6. a** BIRC6 was immuno-precipitated from the HEK293T cell lysates by the anti-BIRC6 antibody or from the mixture of the purified His-LC3 and FLAG-BIRC6 by anti-FLAG, following supplementation of the LIR1 peptide (NPQTSSFLQVLV) or its mutant (NPQTSSALQALV) at 100 μM. Protein levels were analyzed by immunoblotting. Asterisks represent IgG light chains from anti-BIRC6 antibodies. The experiment was repeated twice independently with similar results. **b** 0.5 mg of the purified His-LC3 were incubated with the purified FLAG-BIRC6 at 0.1 μM in 1.2 mL of elution buffer in the absence or the presence of 100 μM LIR1 peptides at RT for 30 min in vitro. The reaction system was passed through Superose 6 Column equilibrated with PBS plus 100 mM KCl. The fraction at 0.5 mL/tube was collected starting from 7 mL, and tubes 10–15 and 21 were collected for immunoblotting following SDS-PAGE. The experiment was repeated twice independently with similar results. **c–e** BIRC6 was immunoprecipitated from the lysates of HEK293T transfected with the FLAG-procasp9-4M by the anti-BIRC6 or anti-XIAP antibody, following supplementation of LIR1

(NPQTSSFLQVLV), hu-LIR (EPLDFDWEIVLEEEM), AVPI, NFEKL at 100 μM, or the related mutant LIR peptides. Protein levels were analyzed by immunoblotting (**c**). Relative levels of co-immunoprecipitated LC3-I (**d**) and LC3-II (**e**) were quantified by densitometry (normalized to LC3-I/LC3-II in Input) with two independent biological replicates. **f, g** The purified FLAG-BIRC6 was incubated with the purified His-LC3 in the presence of the different concentrations of LIR1 (NPQTSSFLQVLV). Immunoblotting was performed following immunoprecipitation with anti-FLAG antibody (**f**). Relative levels of co-immunoprecipitated His-LC3 were quantified by densitometry (normalized to 0 μM LIR1) with two independent biological replicates (**g**). **h** The model mechanism by which LIR1 and hu-LIR peptides inhibit the binding of LC3 to BIRC6 dimer. Gray arc in one BIRC6 molecule represents LIR (aa 4670–4681), and red arc in the other anti-parallel BIRC6 molecule represents the LIR-pairing region. I: LC3-I, II: LC3-II. Source data for (**a–g**) are provided as a Source Data file.

---

defocus range of −2 to −3 μm. The images were recorded using a Gatan GIF K3 camera at a super-resolution counting mode with a dose rate of 17.8 e/s/Å2 and a total exposure time of 3.84 s, and at a movie recording mode with 32 frames for each micrograph.

## Image processing

7331 movie stacks were collected for the BIRC6 samples and pre-preprocessed in RELION4.0[58]. Local drift correction, electron dose weighting, and 2-fold binning were applied to the super-resolution movie stacks using MotionCor2[59], generating the summed micrographs in both dose-weighted and not dose-weighted versions. The unweighted micrographs were used in the steps of CTF estimation with CTFFIND4[60], particle picking, and initial 2D and 3D classification, while the dose-weighted micrographs were used for fine 3D classification and final refinement.

Around 1000 micrographs were first selected and subjected to several rounds of manual and auto-particle picking, 2D and 3D classifications, using both RELION and CryoSPARC[61], to produce the accurate 2D averages for auto-particle picking. With these 2D averages, template matching auto-picking was applied to the complete dataset, with 1925 K particles picked. After two rounds of 3D classification, 72 K particles were kept, with their 3D density maps showing fine structural features (Supplementary Fig. 2c). To recover more particles from the dataset, Topaz[62] was further used to pick particles employing the 72 K good particles as the training dataset, resulting in a set of 1583 K particles (Supplementary Fig. 2d). Similarly, after two rounds of 3D classification, 95 K good particles were kept. To avoid some good particles were mistakenly sorted into the discarded groups during classification, the second round of 3D classification for both particles sets from template matching and Topaz auto-picking was repeated, but in a supervised method, and with the two sets of density maps from the two-particle sets serving as the references in a reciprocal manner (Supplementary Fig. 2c, d). Finally, after pooling the four sets of good particles and removing duplicates, a final set of 154 K particles was subjected to CTF refinement, Bayesian polishing, and C2 symmetry imposed 3D refinement, generating a final global density map at 3.6 Å resolution (Supplementary Fig. 2e, k). The resolution of the BIRC6 core region was improved to 3.5 Å by a round of focused refinement (Supplementary Fig. 2f).

Since the local resolutions of the two ends of ArmRD were relatively lower than the central region, the density maps were further refined focused on one-half of the U-shaped structure with C2 symmetry expanded particles (Supplementary Fig. 2g). As a result, a significant improvement in structural details was observed for both the ends of ArmRD. On the basis of these optimized density maps, the local density of the N-terminal region was further optimized by a skip-alignment-focused 3D classification, which showed the dynamic nature of the N-terminal region (Supplementary Fig. 2h). One group of particles (45 K) whose local density map presented improved

structural details, was subjected to further refinement (Supplementary Fig. 2h).

A composite map was created using phenix.combine_focused_maps[63] (Supplementary Fig. 2l), with the density of residues 31–943 originated from the locally optimized map of the N-terminal region, with residues 984–1419, residues 1531–1876 and residues 4139–4485 from the optimized half map, and residues 1420–1516 and residues 1894–4075 from the optimized map of BIRC6 core region. The local resolution of the composite map was calculated using the built-in local resolution estimation tool of RELION based on the two composite half maps, which were also created using phenix.combine_focused_maps, and displayed using ChimeraX[64] (Supplementary Fig. 2l).

The residual density in the central cavity was also optimized on the basis of the global map to identify potential substrate. With C2 symmetry expanded particles, a skip-alignment focused 3D classification separated two groups of particles with Smac docked in the central cavity. In one group, Smac leans to the left side of the cavity while in another group Smac leans to the right side (Supplementary Fig. 2i). The binding modes of Smac to BIRC6 in the structures of the two groups were similar, thus their sorting into two groups might be due to the global C2 symmetry of the BIRC6 dimer. To merge the two groups of particles, these particles were further processed by a round of supervised 3D classification after duplicate removal and C2 symmetry expansion (Supplementary Fig. 2i). All the Smac-bound particles were then subjected to a global refinement, yielding a density map at 4.7 Å resolution. In the groups of particles with other than Smac, residual densities could also be found in the central cavity (Supplementary Fig. 2i). Accordingly, some other substrates of BIRC6, including caspases 3, 6, 7 and 14, HtrA2 and LC3, and interacting proteins, such as UBA6 and Nrdp1, were also detected in mass spectrometry (Supplementary Fig. 1e and Supplementary Data 1). However these densities couldn't be well assigned due to the resolution limit.

## Model building

The initial model for model building was predicted using Alphafold2[65]. The sequence of mouse BIRC6 was divided into two overlapped regions (residues 1–2588 and residues 1011–4845), and subjected to model prediction using locally installed Alphafold2. The predicted models were separated into domains or secondary structural motifs, and manually fitted into the density map using Chimera[66]. Model adjustment or rebuilding was next performed using COOT[67]. The model was further refined against the composite global map of BIRC6 dimer using phenix.real_space_refinement[63] with secondary structural, NCS (non-crystallographic symmetry), and geometry constraints applied, to optimize the overall geometry quality.

To reveal the interaction between the N1 region (residues 27–358) of BIRC6 and the LIR1 peptide (NPQTSSFLQVLV), the structure of the complex of the whole N-terminal section of BIRC6 (residues 1–943, including both the N1 and N2 regions) and LIR1 was predicted using

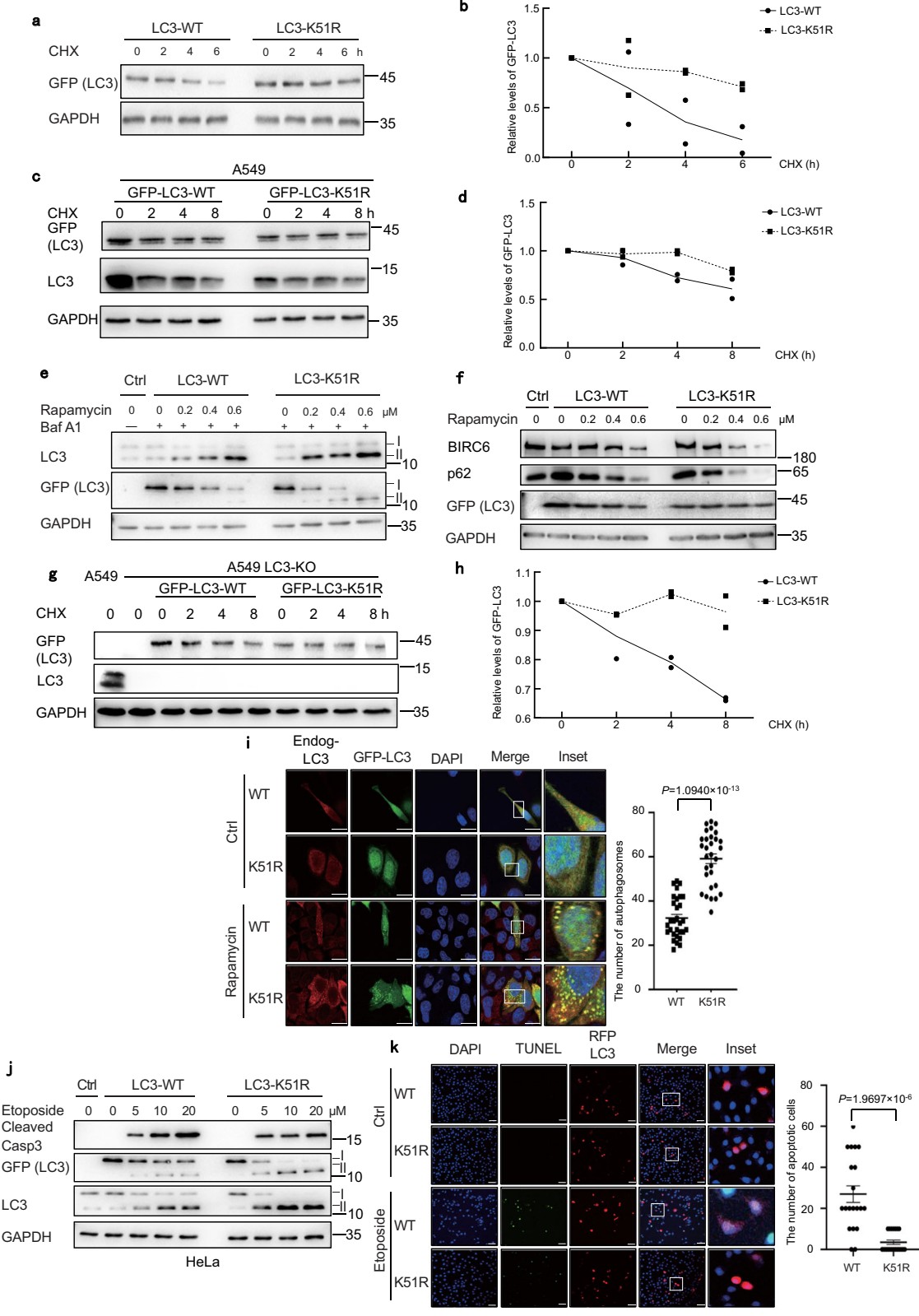

AlphaFold2 with the option "multimer". All the 25 generated models showed a consistent interaction interface between LIR1 and the N1 region, and no interaction between LIR1 and the N2 region was predicted. The top-ranking model was depicted in Supplementary Fig. 5f.

## Cell culture

The wild-type and the LC3-deficient human lung adenocarcinoma A549 cells were kind gifts from Prof. Qinfang Liu. The wild-type and the Atg5-

deficient mouse embryonic fibroblast (MEF) cells were kindly provided by Profs. Y. Ohsumi and A. L. Goldberg. HEK293F cells (Gibco, #11625019) were cultured in SMM 293-TII (HEK293 Cell Complete Medium Animal Free, Serum Free, Sino Biological Inc.). HEK293F cells were kept at 37°C with 5% $CO_2$ and incubation for 120 rpm. MDA-MB-453 (ATCC HTB-131), A549 (ATCC CRM-CCL-185), 293 T (ATCC CRL-3216), HeLa (ATCC CRM-CCL-2), and COS7 (ATCC CRL-1651) cells were cultured in Dulbecco's modified Eagle's medium (DMEM) supplemented with 10%

**Fig. 6 | Mutation of LC3 ubiquitylation site promotes autophagy, but suppresses apoptosis. a–d** HEK293T (**a**) or A549 (**c**) cells were transfected with GFP-LC3-WT or LC3-K51R and then treated with 100 μg/mL CHX. Protein levels were analyzed by immunoblotting, and quantified by densitometry (normalized to GAPDH) with two independent biological replicates in HEK293T (**b**) or A549 (**d**). **e** HEK293T cells were transfected with GFP-LC3-WT or LC3-K51R and then treated with rapamycin for 24 h and/or 0.1 μM Bafilomycin A1 for 2 h. Protein levels were analyzed by immunoblotting with two independent biological replicates. **f** HEK293T cells were transfected with GFP-LC3-WT or LC3-K51R and then treated with rapamycin for 24 h. Protein levels were analyzed by immunoblotting with two independent biological replicates. **g, h** The LC3-deficient A549 (LC3-KO) cells were transfected with GFP-LC3-WT or GFP-LC3-K51R, and then treated with 100 μg/mL CHX. Protein levels were analyzed by immunoblotting (**g**), and quantified by densitometry (normalized to GAPDH) with two independent biological replicates (**h**).

**i** HeLa cells were transfected with GFP-LC3-WT or LC3-K51R and then treated with 0.5 μM rapamycin for 24 h. LC3 puncta were detected under a confocal microscope following immunostaining. *n* = 30 images, Scale bar: 10 μm. When GFP-LC3-K51R was transfected, many puncta are still green after merging, probably because the antibody staining is not efficient. **j** HeLa cells were transfected with GFP-LC3-WT or LC3-K51R and then treated with etoposide for 24 h. Protein levels were analyzed by immunoblotting with two independent biological replicates. **k** HeLa cells were transfected with RFP-LC3-WT or LC3-K51R and then treated with 20 μM etoposide for 12 h. The numbers of apoptotic cells were detected by TUNEL assay under a confocal microscope. *n* = 20 images, Scale bar: 10 μm. The white rectangles indicate the enlarged areas. The experiments (**i, k**) were performed with three replicates and are shown as mean ± standard deviation, two-way ANOVA. I: LC3-I, II: LC3-II. Source data are provided as a Source Data file.

## Immunoblotting and antibodies

The cell lysis for caspases was carried by sonication in lysis buffer (20 mM Hepes-KOH, pH 7.5, 10 mM KCl, 1.5 mM $MgCl_2$, 1 mM EDTA, 1 mM EGTA, 1 mM DTT) with the protease inhibitor mixture (Roche, #4693132001) after treatment by starvation in the HBSS starvation medium with 1 g/L glucose, but without serum or amino acids, for up to 6 h or by rapamycin. For other proteins, the lysis buffer contained 20 mM Tris-HCl pH 8.0, 500 mM NaCl, 1% NP-40, 10% glycerol, and 1 mM $ZnCl_2$. Equal amounts of protein samples were separated by SDS-PAGE and transferred onto PVDF membranes. Antibodies or antisera against the BIRC6 (1:1000, BD Biosciences, #611193), XIAP (1:1000, BD Biosciences, #610717), Smac (1:1000, Zymed, #35-6600), HtrA2 (1:1000, oncogene, #Am82), ATG5 (1:1000, Cell Signaling Technology, #12994), Myc (1:3000, Santa Cruz Biotechnology, #sc-40), GFP (1:3000, Absin, #137960), Apaf-1 (1:1000, Cell Signaling Technology, #8969), PA28γ (1:3000, Enzo, #PW8190), GAPDH (1:3000, Santa Cruz Biotechnology, #25778), p62 (1:1000, Cell Signaling Technology, #88588 S), UBA6 (1:1000, Cell Signaling Technology, #13386), FLAG (1:3000, Sigma-Aldrich, #F3165), Caspase 9 (1:1000, Cell Signaling Technology, #9508), Caspase 3 (1:1000, Cell Signaling Technology, #9662), Cleaved Caspase 3 (1:1000, abcam, ab32042), Caspase 7 (1:1000, Sigma-Aldrich, #SAB4503316), LC3 (1:5000, Sigma-Aldrich, #L7543), β-actin (1:5000, Sigma-Aldrich, #A5441) antibodies were used as primary antibodies to detect the corresponding proteins. Peroxidase-conjugated anti-mouse IgG (1:5000, ZSGB-BIO, #ZB-5305), anti-rat IgG (1:4000, ZSGB-BIO, #ZB-2307) or anti-rabbit IgG (1:3000, ZSGB-BIO, #ZB-5301) was used as the secondary antibody. The protein bands were visualized by an ECL detection system (EMD Millipore).

fetal bovine serum (FBS), 100 U/mL penicillin and 100 μg/mL streptomycin. MEF cells were cultured in the above medium with addition of 1% non-essential amino acids and 200 μM β-mercaptoethanol. All cells were kept at 37 °C with 5% $CO_2$. Cell starvation was performed by incubating cells in HBSS medium at 37 °C with 5% $CO_2$ after the PBS wash twice. Rapamycin (Sigma-Aldrich, #V900930) was used in the normal DMEM medium at the indicated concentration for 24 h. Bafilomycin A1 (Enzo life sciences) was used at a final concentration of 10 nM in HBSS medium for the time points indicated.

## Plasmids and transfection

Myc-tagged full-length Smac (residues 1–239) with a Myc-His6 tag at the C terminus was subcloned into pcDNA6 (Invitrogen)as described previously[12]. Myc or FLAG-tagged procaspase 9 plasmid was prepared by cloning the procaspase 9 gene into a pcDNA-based vector (Invitrogen). Procaspase 9 with three mutations (Asp315Ala, Asp330Ala, and Glu306Ala; Procasp9-3M) or with 4 mutations (E306A–D315A–D330A–C287A; Procasp9-4M) was generated using a direct mutagenesis strategy. 293 T cells were transfected with plasmids using Lipofectamine 2000 according to the manufacturer's protocol (Life Technologies, Inc.). His-tagged procaspase 9–4 M (E306A–D315A–D330A–C287A; His-Procasp9-4M) was prepared by cloning the FLAG-Procasp9-4M into a pET-based vector (Invitrogen). His-tagged BIRC6-N1 (residues 27–358) and His-tagged BIRC6-N1 (residues 358–543) were prepared by cloning the FLAG-BIRC6 into the pET-based vector (Invitrogen).

## Confocal immunofluorescence microscopy

Cells grown on glass coverslips in 6-wells were fixed in 4% polyformaldehyde for 10 min at room temperature, followed by membrane permeation using 100 μg/mL digitonin in PBS for 5 min. Cells were blocked in 3% BSA before primary antibodies were applied. Anti-procaspase 9 mouse IgG (1:100, Santa Cruz, #SC56076) and anti-LC3 rabbit IgG (1:100, MBL, #PM036) were incubated in a moist container for 1 h at room temperature. Then, the secondary anti-mouse antibodies conjugated with FITC and anti-rabbit with Alex-594 (ZSGB-BIO, #ZF-0513) were applied to the cells in a moist container for 1 h at room temperature. The DNAs in nuclei were stained with 4′-6-diamidino-2-phenylindole (DAPI, Sigma-Aldrich, #28718-90-3) at the final preparation step. The slides were viewed on a Zeiss LSM 700 confocal microscope using a ×100 oil objective and laser lines at 488 nm and 555 nm for excitation.

## Immunoprecipitation

Cells were lysed with the lysis buffer as in the immunoblotting procedure, and the lysates were immunoprecipitated with specific antibody and protein A/G-Sepharose (Sigma-Aldrich) at 4 °C for 2 h. The precipitants were washed three times with the lysis buffer, and then eluted with the sample buffer containing 1× SDS at 97 °C for 6 min.

## Apoptosis assays

Apoptotic DNA fragmentation was analyzed using the DeadEnd™ Fluorometric TUNEL System according to the standard paraffin-embedded tissue section protocol (Promega). The DNAs in nuclei were stained with DAPI at the final preparation step. The slides were viewed on a Zeiss LSM 700 confocal microscope using a ×100 oil objective and laser at 488, 555, or 639 nm for excitation.

## Inhibition of caspase activity assay

To activate caspases in cell lysates, 160 μg of 293 T cell extracts in each reaction (100 μL) were activated by cytochrome c and dATP essentially as described[12]. The purified FLAG-tagged BIRC6 was added to the reaction. The caspase activity was analyzed in the buffer containing 0.1 M Tris, pH 6.8, 10 mM dithiothreitol, 0.1% CHAPS, 10% polyethylene glycol, and 20 μM of the caspase 9 substrate Ac-LEHD-amino-4-methylcoumarin-AMC (Enzo, #ALX-260-080-M001) or caspase 3 substrate Z-DEVD-7-AMC (Merck, #235425). The release of AMC from the substrates was monitored continuously at 380/460 nm (excitation/emission) at 30 °C for up to 2 h. Initial rates were derived by calculating the gradient of the linear part of the curve. The linear part of the curve (non-plateau period) was intercepted, and the rates of the treatment groups with different BIRC6 concentrations were calculated using

(RFU2-RFU1)/t. The rates of the 0 nM treatment group were represented by Vi, and the rates of the other treatment groups with different BIRC6 concentrations were represented by V0, with inhibition ratio =(Vi-V0)/Vi. The curve was fitted by GraphPad Prism 8.0.1 software.

## Quantification and statistical analysis

Unless stated otherwise, significance levels for comparisons between 2 groups were determined by one-way ANOVA. Two-way repeated ANOVA was employed to compare multiple repeated measurements among groups. Data are reported as mean ± standard deviation from three independent biological replicates, normal distribution. All images were chosen at random while blinded, and were quantitated using Image J.

## Data availability

The data that support the findings of this study are available from the authors upon request. The Cryo-EM density maps of the global BIRC6, the composite map, the core region of BIRC6, the half map of the core region, the N-terminal region of BIRC6, and the Smac-BIRC6 complex have been deposited in the Electron Microscopy Data Bank (EMDB) with accession codes EMD-35759, EMD-38464, EMD-38461, EMD-38462, EMD-35758, and EMD-35760. The atomic models of the global BIRC6 have been deposited in the Protein Data Bank (PDB) under PDB ID 8IVQ. The raw mass-spectrometric data generated in this study have been deposited in the ProteomeXchange Datasets database with accession code PXD042246. Source data are provided with this paper.

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

## Acknowledgements

We are grateful to Professor Y. Ohsumi and A.L. Goldberg for kindly providing *Atg5*^+/+ and *Atg5*^−/− MEF cells and Professor Qinfang Liu for LC3^+/+ and LC3^−/− A549 cells. We thank the Cryo-EM platform of Peking University for the Cryo-EM grid screening and data collection, the High-performance Computing Platform of Peking University for the computation, and the National Center for Protein Sciences at Peking University for negative-staining EM and mass spectrometry analyses. This study was supported by the National Key R&D Program of China (2019YFA0802100) and the National Science Foundation of China (32330027 and 82350006) to X.B.Q., the National Key R&D Program of China (2019YFA0508904) to N.G., the National Science Foundation of China (31922036 and 32271257) to N.N.L., and Beijing Municipal Natural Science Foundation (5202014) and Tang scholarship to T.X.J.

## Author contributions

S.S.L., T.X.J., F.B., J.L.Z., and G.F.W. devised and performed the majority of experiments, and analyzed data with the assistance of G.H.Y., J.Y.K, Y.F.Q., L.B.F., and P.W.; X.B.Q., N.G., and N.N.L. conceived the project, supervised the experiments, analyzed data, and wrote the manuscript with input from all authors.

## Competing interests

The authors declare no competing interests.
