## [Peer Review File · Nature Communications]

Molecular mechanisms underlying the BIRC6-mediated regulation of apoptosis and autophagyREVIEWER COMMENTS

Reviewer #1 (Remarks to the Author):

BIRC6 is a large protein of the IAP family and regulates apoptosis and autophagy. Despite its size (530 kDa), previously only 2 small domains were annotated, and the structural basis for the functions of BIRC6 was unresolved. The manuscript by Liu et al., "Structural basis for BIRC6 to balance apoptosis and autophagy" aims to provide the structural basis for how BIRC6 regulates the balance between apoptosis and autophagy. Different to other IAPs, BIRC6 only contains a single BIR domain and is a E2/E3 hybrid enzyme, containing a C-terminal UBC domain.

In the first part of the manuscript, the authors purify overexpressed FLAG-tagged mouse BIRC6 from HEK293F cells and solve its cryo-EM structure at 3.6 Å resolution. The structure shows a well-defined antiparallel U-shaped homodimer of two BIRC6 molecules and allows identification of many previously unannotated domains, most prominently 31 armadillo repeats that form the scaffold for the other functional domains of BIRC6. The structural arrangement likely brings the BIR and UBC domains (the latter is flexible and not resolved in the structure) into proximity, pointing to a mechanism for BIRC6's catalytic activity. The authors further use focused classification and refinement to identify additional density in the central cavity of the U-shape and assign this to the BIRC6 inhibitor SMAC that was co-purified from the expression system as shown by mass spectrometry.

This structural aspect of the manuscript is generally well-supported by the presented data, but no cross-validation of the Smac/BIRC6 interactions have been performed using orthogonal methods. The BIRC6/Smac structure agrees with three other cryo-EM structures reported as pre-prints on bioRxiv in the last ~ 5 months (Hunkeler et al.; Ehrmann et al.; Dietz et al.). While some of these preprints report additional BIRC6 structures (e.g., complexes with substrates caspase-3, HtrA2), they do not take away from this aspect of the manuscript presented here.

In the second part of the manuscript, the authors investigate the relationship between apoptosis and autophagy and how these are differentially regulated by BIRC6. The authors focus on Smac, caspase-9, caspase-3 and LC3B, and their interactions with BIRC6. This part of the study overall is less well refined than the first part, and several conclusions lack sufficient experimental evidence as outlined in detail below.

In summary, the work presents an interesting and highly anticipated structure of BIRC6, but the additional work is less well supported and is disconnected from the structural work, therefore, in our view, the work does not provide the structural basis for how BIRC6 balances apoptosis and autophagy as stated prominently in the title and abstract.

In our view, the study would need substantial additional work, clarifications, and restructuring before publication.

We summarise additional technical issues below.

Major points:

I) The interactions between Smac and BIRC6 observed in the structure should be validated using orthogonal assays, e.g., by testing various interaction site mutants in functional assays or biophysical binding assays.

II) Caspase activity assay in Fig. ED4c,d and Fig. 2m: The presented data in Fig ED4c,d does not support the conclusion that "the purified BIRC6 could strongly inhibit the activity of caspase 9" (p. 10). I would expect the fluorescence (RFU) increase over time and inhibition by BIRC6 result in a change in the initial rate of fluorescence release. Instead, the signal fluctuates around a constant value over the duration of the experiment, and higher BIRC6 concentrations lead to increased RFU signal. Further it is unclear how the graph in Fig 2m is derived from the raw data in Fig ED4c,d, i.e. how was % inhibition calculated, and at which time point? The figure legend states that the graph in Fig 2m shows a single representative of three independent biological replicates. Data

points for all 3 replicates should be presented.

III) Can the authors better explain the experimental results of their experiments with the different LIR peptides (BIRC6-LIR and hu-LIR) on p. 12/13? Why do the two peptides show opposite effects on the BIRC6-LC3B interaction (Fig. 3f)? The proposed mechanism (opening of the BIRC6) dimer is highly speculative and needs to be supported by additional data. Further, the LIR sequence is located within the UBC domain, which based on the structure is highly dynamic and therefore presumably not interacting in stable dimer interactions, making the proposed model less likely.

IV) The proposed interaction between the BIRC6 UBL domain and UBA6 is very speculative and should be removed or supported by actual data, at least pull-downs or biophysical binding studies with e.g., a BIRC6 Δ UBL mutant

V) Fig ED4a: The presented gel does not convincingly show co-elution of BIRC6 and Casp9-4M. Only a minor fraction of His-Casp 9-4M overlaps with the tail end of the FLAG-BIRC6 peak. A control of His-Casp 9-4M in the absence of BIRC6 is required. The elution profiles of the proteins should also be shown.

VI) Some of the blots and gels are of very poor resolution and quality, making it impossible to validate the corresponding conclusions derived by the authors. The worst examples include Fig. 2l, BIRC6 and XIAP panels, all panels of Fig 3e and Fig ED5d (containing image processing artifacts). These experiments should be repeated, and better-quality blots presented.

VII) Fig2k: What is sample "C"? Why is much more myc(BIRC6) present in that sample?

VIII) Fig ED1d: the legend describes 3 FLAG elutions, but the gel seems to come from a SEC experiment. Please correct/clarify this.

IX) Fig ED1e: The legend describes that other interactors are provided as online materials. Please provide this list.

X) Fig ED4: The legend does not match the figure.

XI) Fig ED5c/d: The quality of the gel in ED5d is very poor and shows some image processing artifacts, making it very difficult to observe the described faint LC3B band in the red box. Can the authors provide a Western blot to better visualise LC3B? What concentration of LIR peptide was used in this experiment?

XII) Fig ED6a,b: Why is the band of the PA28y weaker at later time points in LC3-K51R but not LC3-WT samples?

XIII) The authors claim that UBA6 binds Ub and therefore may bind the BIRC6 UBL domain. The authors should provide a reference for UBA6 binding to Ub.

XIV) Details about the Cryo-EM data collection are missing, Table ED1 states "Triton X-100 treated sample". Neither the methods nor the main text contain information about this treatment.

XV) Methods for the mass spectrometry analysis need to be included.

XVI) The final PDB validation report rather than the preliminary report that is not suited for manuscript review should be submitted.

XVII) Fig. 2 covers two very different sets of experiments and should probably be split into 2 figures, 2a-j and 2k-l.

XVIII) p. 10: "Moreover, the Smac with IBM mutations, where the AVPI stretch was mutated into AAAI, further reduced the co-immunoprecipitation of either BIRC6 or XIAP with HtrA2 (Fig. 2l), suggesting that the region other than the IBM of HtrA2 can also mediate its association with BIRC6." It is not clear how the authors come to this conclusion based on the presented data. Can the authors please expand on this.

Minor points:

i. Fig 1a: the domain abbreviations should be defined in the figure legends.

ii. p. 11: data in Fig 3a and ED5a does not show HEK293T cells as stated in the text.

iii. P.15: "because loss of LC3 ubiquitylation did the opposite as shown in this study (Extended Data Fig. S6d)". It is unclear which studies are compared here and what "the opposite" is referring to.

iv. p. 2: please specify what is meant by "exotic pathogens" or rephrase.

v. Fig 3e,f: The figure labels with lots of + and - are very difficult to follow, further confounded by the fact that the labels do not align with the lanes of the gel. This is also evident in other figures.

vi. Fig 4d/f: Y-axes labels are missing.

vii. Fig 4f: The figure legend states n = 30 images, but only 20 data points are shown for each sample in the graph.

viii. Fig ED1b: The presented SEC profile does not show if BIRC6 is a dimer or not as no calibration is shown. The figure legend describes two different experiments using normal and cross-linked

protein. Please specify which sample is used for the chromatogram shown.

ix. The description "tube" for the gel samples in the SEC experiments (e.g., Fig. ED1c) does not provide any useful information. Strongly suggest providing the elution volumes.

x. Fig ED1c: Typo in label: FLAG-BIRC6

xi. Almost all gels and blots lack molecular weight markers (e.g., Fig. ED 1c,d)

xii. Main text and Fig. ED2g "N-terminal session". Do the authors mean "N-terminal section"?

xiii. Fig ED3 i,j: please specify that CBM32 in i) is BIRC6 and that CBM32 is from another protein; which one? Also show reference not just the PDB ID. Same for the PDB ID in k.

xiv. Fig ED6c: label myc should be FLAG based on the figure legend.

xv. Page 9, penultimate paragraph: The Fig ED2i should be referenced instead of Fig ED2h.

xvi. Page 15, last line – 'Extended Data Fig. S6d' – remove the 'S'.

xvii. Page 17, line 19 – Cryo-EM is misspelled as cry-EM.

Reviewer #2 (Remarks to the Author):

The authors determined the cryo-EM structure of BIRC6, a gigantic 530 kDa protein with little information about its structure known previously. BIRC6 has also been noted to interact with both autophagy and apoptosis pathway. In this manuscript, the authors revealed the cryo-EM structure of BIRC6 at 3.6Å. The authors found that BIRC6 forms a symmetric homodimer with multiple domains in a U-shape structure.

The data on the structural interaction of Smac with BIRC6 was clear. Smac, a previously known ubiquitination substrate of BIRC6, was copurified in the BIRC6 samples, which allowed the authors to elucidate the binding of Smac with BIRC6 in dimeric conformation. Unfortunately, the remaining parts of the manuscript are very unclear and sometimes, digressed far from the first part and with conclusions unjustified or unclear.

The major issues:

1) The authors also tried to elucidate the structural information of BIRC6 interaction with caspase-9, which was shown previously by the authors. Since the authors have previously reported that BIRC6 can bind with purified processed caspase-9, it is unclear why the authors decided to investigate the binding of unprocessed caspase-9 and BIRC6 in this structural manuscript. It is also unclear why the authors continued to have a section on the interaction of XIAP with procaspase-3, since this is a BIRC6 paper.

2) Then the manuscript digressed further into autophagic regulation of caspase-9 levels. It is doubtful that the levels of caspase-9 change in rapamycin treated cells can affect apoptosis (not shown). It is also not clear if the changes in caspase-9 with rapamycin is regulated by BIRC6 (not shown). Thus, it is a big overstatement to say that BIRC6 regulates the balance between autophagy and apoptosis.

3) LC3-K51R may have an effect on its expression level or stability, but it is not clear how is that related to BIRC6.

Specifics:

1. The title "Structural basis for BIRC6 to balance apoptosis and autophagy" should be changed. First, it is unclear if apoptosis and autophagy is in any kind of balance or needs to be balanced. Apoptosis promotes cell death; while autophagy promotes degradation and survival. Second, even if there is a balance between apoptosis and autophagy, it is not likely that BIRC6 should be the key regulator as the strongest target of BIRC6 seems to be Smac, which serves a minor role in apoptosis anyway. Third, the authors did not provide data to convincingly demonstrate this point in the manuscript.

2. The subtitle "LC3 ubiquitination promotes apoptosis" is not justified and should be changed.

Reviewer #3 (Remarks to the Author):

In this study, the authors reveal an excellent cryo-EM structure of mouse BIRC6 at a global resolution of 3.6Å. This structure is organized by two BIRC6 molecules, which arrange in an antiparallel manner to form a U-shaped structure. Based on this structure, authors analyze the organization of different regions/motifs of BIRC6 and the potential interaction fashion of BIRC6 and SMAC. Next, authors demonstrate that procaspase-9 is degraded by autophagy. At the end of this study, authors show that overexpression of LC3 K51R mutant promotes autophagy and inhibits apoptosis. This work provides important insights into how BIRC6 regulates autophagy and apoptosis.

My biggest concern is the disconnect between structural and functional research. Authors illustrate an excellent structure of this giant protein BIRC6, however, this structure is not connected with the function of BIRC6 in regulating autophagy or apoptosis as authors show in the other part of this study. For instance, in page 13, authors claim that LIR1 peptide might change the conformation of BIRC6 LIR, in page 16, authors indicate BIRC6 LIR does not overlap with the binding regions of SMAC. The related structures would be shown to support these hypotheses. Another major concern is the reproducibility. The quantification of WB and IFA from multiple repeated experiments should be provided.

Other specific comments:

- 1: Figure 2k, BIRC6 is shown to interact with procaspase-9. Does cleaved Cas-9 interact with BIRC6? How does Smac regulate this interaction?
2. The data of Figure 2a-d show the autophagic degradation of procaspase-9, however, Figure 2e and 2f is about the function of BIRC6 LIR. These are two unrelated contents. Figure 2 should be split to two figures.
3. Authors conclude that BIRC6 binds LC3 through LIR. As shown in figure 2e,2f. the putative BIRC6 LIR peptide LIR1 is not able to compete BIRC6 for binding LC3, but a hypothetical universal LIR peptide indeed reduces the interaction of BIRC6 and LC3. This result indicates the putative BIRC6 LIR is not functional. The conclusion needs reconsideration.
4. Figure 4. LC3 null cells need to be used to eliminate any effect of endogenous LC3.
5. Figure 4. Lysine 51 of LC3 is not only ubiquitinated, but also acetylated. Besides, K51R mutant loses the interaction with autophagy receptor, e.g. p62. Thus, the observed effects by K51R overexpression in this Figure should not be simply attributed to loss of ubiquitination. Authors should exclude other possibilities or the conclusion needs reconsideration.
6. EMD and PDB number should be provided.
7. A bioRxiv preprint about BIRC6 structure has been cited in this article. Two more similar preprints, which are posted in last August by Moritz Hunkeler and Larissa Dietz, should also be cited and discussed.

Re: Manuscript NCOMMS-22-53213, “Structural basis for BIRC6 to balance apoptosis and autophagy” by Liu et al.

Point-to-point responses to reviewers

Reviewer 1:

BIRC6 is a large protein of the IAP family and regulates apoptosis and autophagy. Despite its size (530 kDa), previously only 2 small domains were annotated, and the structural basis for the functions of BIRC6 was unresolved. The manuscript by Liu et al., “Structural basis for BIRC6 to balance apoptosis and autophagy” aims to provide the structural basis for how BIRC6 regulates the balance between apoptosis and autophagy. Different to other IAPs, BIRC6 only contains a single BIR domain and is a E2/E3 hybrid enzyme, containing a C-terminal UBC domain. In the first part of the manuscript, the authors purify overexpressed FLAG-tagged mouse BIRC6 from HEK293F cells and solve its cryo-EM structure at 3.6 Å resolution. The structure shows a well-defined antiparallel U-shaped homodimer of two BIRC6 molecules and allows identification of many previously unannotated domains, most prominently 31 armadillo repeats that form the scaffold for the other functional domains of BIRC6. The structural arrangement likely brings the BIR and UBC domains (the latter is flexible and not resolved in the structure) into proximity, pointing to a mechanism for BIRC6’s catalytic activity. The authors further use focused classification and refinement to identify additional density in the central cavity of the U-shape and assign this to the BIRC6 inhibitor SMAC that was co-purified from the expression system as shown by mass spectrometry.

This structural aspect of the manuscript is generally well-supported by the presented data, but no cross-validation of the Smac/BIRC6 interactions have been performed using orthogonal methods. The BIRC6/Smac structure agrees with three other cryo-EM structures reported as pre-prints on bioRxiv in the last ~ 5 months (Hunkeler et al.; Ehrmann et al.; Dietz et al.). While some of these preprints report additional BIRC6 structures (e.g., complexes with substrates caspase-3, HtrA2), they do not take away from this aspect of the manuscript presented here.

In the second part of the manuscript, the authors investigate the relationship between apoptosis and autophagy and how these are differentially regulated by BIRC6. The authors focus on Smac, caspase-9, caspase-3 and LC3B, and their interactions with BIRC6.

This part of the study overall is less well refined than the first part, and several conclusions lack sufficient experimental evidence as outlined in detail below.

In summary, the work presents an interesting and highly anticipated structure of BIRC6,

but the additional work is less well supported and is disconnected from the structural work, therefore, in our view, the work does not provide the structural basis for how BIRC6 balances apoptosis and autophagy as stated prominently in the title and abstract.

In our view, the study would need substantial additional work, clarifications, and restructuring before publication.

Responses: *We thank the reviewer for praising our work on the structure of BIRC6 interesting and highly anticipated, and agree with the reviewer that the functional part is less well refined. BIRC6 has previously been shown to be important in the regulation of both apoptosis and autophagy (Qiu et al., EMBO J 23: 800, 2004; Jiang et al., PNAS 116: 13404, 2019). In the revised version, we have provided additional data and re-structured the manuscript to strengthen our conclusions as advised by the reviewer. For example, we have now performed additional experiments to show that (1) Smac regulates the interaction between the active caspase 9 and BIRC6 (revised Fig. 3c) (providing an explanation how structural differences between procaspase 9 and active caspase 9 influence their activities and functions), (2) high concentrations of LIR1 peptides can also outcompete LC3 for binding the LIR motif of BIRC6 (revised Fig. 5d) (providing a link between LIR motif in BIRC6 structure and its function in regulating LC3 as well as autophagy), and (3) the levels of procaspase-9 and BIRC6 are reduced at the same times during autophagy (revised Fig. 4d-e) (strengthening a role of BIRC6 in regulating both apoptosis and autophagy).*

Major points:

I) The interactions between Smac and BIRC6 observed in the structure should be validated using orthogonal assays, e.g., by testing various interaction site mutants in functional assays or biophysical binding assays.

Responses: *We agree with the reviewer that the interactions between Smac and BIRC6 in the structure should be tested by using various interaction site mutants or biophysical binding assays. When this manuscript was under review, three other groups reported similar structural results on BIRC6-Smac complex in Science, which appeared previously as preprints on bioRxiv as noted by the reviewer (Hunkeler et al., Science 379: 1105, 2023; Ehrmann et al., Science 379: 1117, 2023; Dietz et al., Science 379: 1112, 2023). These groups presented in vitro biophysical binding assays. As a complement, we had confirmed the interactions between Smac and BIRC6 by using the Smac mutant in the cellular system, where the AVPI stretch in Smac was mutated into AAI (revised Fig. 3b). To avoid repeating unnecessarily, we have now cited the reports by others to further support our conclusions in the revised version (Page 21).*

II) Caspase activity assay in Fig. ED4c,d and Fig. 2m: The presented data in Fig ED4c,d does not support the conclusion that “the purified BIRC6 could strongly inhibit the activity of caspase 9” (p. 10). I would expect the fluorescence (RFU) increase over time and inhibition by BIRC6 result in a change in the initial rate of fluorescence release.

Instead, the signal fluctuates around a constant value over the duration of the experiment, and higher BIRC6 concentrations lead to increased RFU signal. Further it is unclear how the graph in Fig 2m is derived from the raw data in Fig ED4c,d, i.e. how was % inhibition calculated, and at which time point? The figure legend states that the graph in Fig 2m shows a single representative of three independent biological replicates. Data points for all 3 replicates should be presented.

Responses: *We thank the reviewer for suggesting us to improve the presentation of the data for the caspase activity assay in Fig. ED4c, d and Fig. 2m. In the revised version, we have added 2 more replicates of the caspase activity assay and provided clearer results in the revised Fig. ED4c-e and Fig. 3d. The conclusion is still same as before.*

III) Can the authors better explain the experimental results of their experiments with the different LIR peptides (BIRC6-LIR and hu-LIR) on p. 12/13? Why do the two peptides show opposite effects on the BIRC6-LC3B interaction (Fig. 3f)? The proposed mechanism (opening of the BIRC6) dimer is highly speculative and needs to be supported by additional data. Further, the LIR sequence is located within the UBC domain, which based on the structure is highly dynamic and therefore presumably not interacting in stable dimer interactions, making the proposed model less likely.

Responses: *We agree with the reviewer that the proposed mechanism (opening of the BIRC6 dimer) is highly speculative. However, it is still tempting to speculate it to work by opening the BIRC6 dimer, especially because the structure around the LIR sequence is highly dynamic so that free LIR-1 peptides are possible to open the dimer for binding LC3. Because the hu-LIR peptide sequence is generally different from that of the BIRC6 LIR region and cannot dimerize with the corresponding BIRC6 region, it is thus free to reach the BIRC6 LIR region and block the LC3 binding, which happens at a low frequency in regularly cultured cells (open status in the revised Fig. 5e). However, BIRC6-LIR1 peptides might competitively and preferentially bind the part of the other BIRC6 molecule, which supposedly dimerizes with the LIR region of BIRC6 (closed status), and then make the BIRC6 LIR region free to bind LC3. If this is the case, further increase of LIR1 peptide concentrations would competitively bind the relaxed BIRC6 LIR and reduce the binding of LC3. To validate this hypothesis, we have now used different concentrations of LIR1 peptides in our co-IP experiments and found that LC3 binds BIRC6 at the highest levels in the presence of 100 μ M of LIR1 peptides, and the binding indeed declines afterwards with the increase of the LIR1 peptide concentrations (revised Fig. 5d). In the revised version, we have now presented a scheme to further explain this model, though it is still speculative (revised Fig. 5e).*

IV) The proposed interaction between the BIRC6 UBL domain and UBA6 is very speculative and should be removed or supported by actual data, at least pull-downs or biophysical binding studies with e.g., a BIRC6 Δ UBL mutant

Responses: *Thanks for the reviewer's suggestion. As introduced in the revised version, unlike UBA1, which binds the C-terminal glycine of ubiquitin only through the thioester bond, UBA6 probably binds ubiquitin not only through the thioester bond,*

but also through non-thioester bonds, though the mechanism remains unclear (Jin et al., Nature, 447: 1135, 2007; Chiu, Sun et al., Mol Cell, 27: 1014, 2007). We have now found that the UBL domain of BIRC6 could co-IP with UBA6 from cell lysates (revised Extended Data Fig. 3b). The previous speculation “raising the possibility that UBA6 directly binds BIRC6 through this ubiquitin-like domain” has changed into “we speculate that UBA6 possibly binds BIRC6 through the UBL domain” in the revised version (Pages 8-9).

V) Fig ED4a: The presented gel does not convincingly show co-elution of BIRC6 and Casp9-4M. Only a minor fraction of His-Casp 9-4M overlaps with the tail end of the FLAG-BIRC6 peak. A control of His-Casp 9-4M in the absence of BIRC6 is required. The elution profiles of the proteins should also be shown.

Responses: As suggested by the reviewer, we have now added the control of His-Casp 9-4M in the absence of BIRC6 (revised Fig. ED4a).

VI) Some of the blots and gels are of very poor resolution and quality, making it impossible to validate the corresponding conclusions derived by the authors. The worst examples include Fig. 2l, BIRC6 and XIAP panels, all panels of Fig 3e and Fig ED5d (containing image processing artifacts). These experiments should be repeated, and better-quality blots presented.

Responses: As suggested by the reviewer, we have now repeated the experiments and provided the clearer results in Fig. 2l (revised Fig. 3b), Fig 3e (revised Fig. 5a) and Fig ED5d (revised Fig 5b). These improved blots or gels still support our previous conclusions.

VII) Fig2k: What is sample “C”? Why is much more myc(BIRC6) present in that sample?

Responses: We are sorry for failing to describe the sample “C”, which was the experimental group transfected only with myc-BIRC6 as shown in the revised Fig. 3a . Because transfection of Procasp9-WT, 3M and 4M at the same time would lead to cleavage of BIRC6, the lane transfected with BIRC6 alone would have more Myc-tagged BIRC6.

VIII) Fig ED1d: the legend describes 3 FLAG elutions, but the gel seems to come from a SEC experiment. Please correct/clarify this.

Responses: As advised by the reviewer, we have repeated the assay and detected the both FLAG-BIRC6 and Myc-Smac by Western blot in the revised Fig. ED1f.

IX) Fig ED1e: The legend describes that other interactors are provided as online materials. Please provide this list.

Responses: As advised by the reviewer, we have now provided the full list of proteins obtained from Mass Spectrometric analyses.

X) Fig ED4: The legend does not match the figure.

Responses: We have now adjusted the legend to match to Fig ED4.

XI) Fig ED5c/d: The quality of the gel in ED5d is very poor and shows some image processing artifacts, making it very difficult to observe the described faint LC3B band in the red box. Can the authors provide a Western blot to better visualise LC3B? What concentration of LIR peptide was used in this experiment?

Responses: Due to file conversion, we are sorry that the faint bands probably were not visible at all for the reviewer. As suggested by the reviewer, we have now repeated the assay and visualized LC3B by Western blot (revised Fig 5b). The concentration of LIR peptide at 100 μ M has now been provided in the revised version.

XII) Fig ED6a,b: Why is the band of the PA28g weaker at later time points in LC3-K51R but not LC3-WT samples?

Responses: As correctly noted by the reviewer, the band of the PA28gamma was weaker at later time points in LC3-K51R, but not LC3-WT samples. It is well established that proteasomes and their subunits can be degraded by autophagy (Marshall et al., Mol Cell 58:1053, 2015). These results might suggest that LC3-K51R also blocked autophagic degradation of PA28gamma. The related explanation has now been included in the revised version (Page 19).

XIII) The authors claim that UBA6 binds Ub and therefore may bind the BIRC6 UBL domain. The authors should provide a reference for UBA6 binding to Ub.

Responses: As suggested by the reviewer, the related references have been provided in the revised version (Chiu et al., Mol Cell 27: 1014, 2007; Jin et al., Nature 447: 1135, 2007).

XIV) Details about the Cryo-EM data collection are missing, Table ED1 states “Triton X-100 treated sample”. Neither the methods nor the main text contain information about this treatment.

Responses: In the previous version, we described “The Cryo-EM data collection” from line 6 to 11 in paragraph 2 of the session “Electron microscopy”. Now, we have separated the session into two parts: “Negative staining EM” and “The Cryo-EM data collection”. Thanks to the reviewer’s comments, we have now corrected the Table ED1 and removed “Triton X-100 treated ” in the revised version.

XV) Methods for the mass spectrometry analysis need to be included.

Responses: As suggested by the reviewer, methods for mass spectrometry have now been included in the revised version (Page 25).

XVI) The final PDB validation report rather than the preliminary report that is not suited for manuscript review should be submitted.

Responses: As suggested by the reviewer, we have now submitted the final PDB validation report.

XVII) Fig. 2 covers two very different sets of experiments and should probably be split into 2 figures, 2a–j and 2k–l.

Responses: *As suggested by the reviewer, we have now split Fig. 2 into figures 2a–j and 3a, b and d.*

XVIII) p. 10: “Moreover, the Smac with IBM mutations, where the AVPI stretch was mutated into AAAI, further reduced the co-immunoprecipitation of either BIRC6 or XIAP with HtrA2 (Fig. 2l), suggesting that the region other than the IBM of HtrA2 can also mediate its association with BIRC6.” It is not clear how the authors come to this conclusion based on the presented data. Can the authors please expand on this.

Responses: *We agree with the reviewer that this statement needs further clarification. As described in the Introduction, cytosolic Smac precursor is proteolytically processed in mitochondria into its mature form, which is released into the cytosol to interact with IAPs with its exposed IAP-binding motif (IBM) in response to apoptotic stimuli (Du et al., Cell 102, 33, 2000; Verhagen et al., Cell 102, 43, 2000). All IBM-containing proteins must be processed to expose their N-terminal IBM for binding regular IAPs, such as XIAP (Srinivasula et al., Nature 410: 112, 2001). However, BIRC6, unlike other IAPs, also binds to the Smac precursor with the unexposed IBM and promotes its degradation in an IBM-independent manner (Qiu and Goldberg, J Biol Chem 280:174, 2005). Although caspase 9, Smac and HtrA2 all share similar IBMs, whether HtrA2 binds BIRC6 in an IBM-independent manner was still unclear. In previous Fig. 2l, we had shown that the Smac with IBM mutations, where the AVPI stretch was mutated into AAAI, further reduced the co-immunoprecipitation of BIRC6 with HtrA2 (previous Fig. 2l, revised Fig. 3b), suggesting that the region other than the IBM of Smac could still compete with HtrA2 for binding BIRC6. Thus, the region other than the IBM of HtrA2 should also mediate its association with BIRC6. As suggested by the reviewer, we have now provided additional explanation for this conclusion in the revised version (Pages 13–14).*

Minor points:

i. Fig 1a: the domain abbreviations should be defined in the figure legends.

Responses: *As suggested by the reviewer, the domain abbreviations have now been defined in the revised Fig 1a legends.*

ii. p. 11: data in Fig 3a and ED5a does not show HEK293T cells as stated in the text.

Responses: *In both cases, MDA-MB-453 cells were used as corrected in the revised version.*

iii. P.15: “because loss of LC3 ubiquitylation did the opposite as shown in this study (Extended Data Fig. S6d)”. It is unclear which studies are compared here and what “the opposite” is referring to.

Responses: *To avoid confusion, we have revised this part into “because mutation of the LC3 ubiquitylation site promoted autophagy and inhibited apoptosis” in the revised version.*

iv. p. 2: please specify what is meant by “exotic pathogens” or rephrase.

Responses: Thanks for the reviewer's critical comment, we have now deleted "exotic" in the revised version.

v. Fig 3e,f: The figure labels with lots of + and – are very difficult to follow, further confounded by the fact that the labels do not align with the lanes of the gel. This is also evident in other figures.

Responses: Thanks for the reviewer's critical comment, we have now improved alignment of figure labels as suggested in the revised version.

vi. Fig 4d/f: Y-axes labels are missing.

Responses: Corrected as suggested.

vii. Fig 4f: The figure legend states $n = 30$ images, but only 20 data points are shown for each sample in the graph.

Responses: Thanks for the correction, and the numbers are now revised accordingly.

viii. Fig ED1b: The presented SEC profile does not show if BIRC6 is a dimer or not as no calibration is shown. The figure legend describes two different experiments using normal and cross-linked protein. Please specify which sample is used for the chromatogram shown.

Responses: As correctly noted by the reviewer, the presented SEC profile in Fig. ED1b did not show whether BIRC6 is a dimer, because we did not expect it until observing it by Cryo-EM. Although we had done the two different experiments, Fig. ED1b just showed the cross-linked proteins as noted in the legend. Since there were three peaks that were mixed, and the peak of FLAG peptide starting from 19 ml was very high at the molecular sieve diagram before crosslinking (data not shown), we only used crosslinked samples for electron microscopy. Now, we have provided results from the Coomassie staining before crosslinking (revised Fig. ED1b).

ix. The description "tube" for the gel samples in the SEC experiments (e.g., Fig. ED1c) does not provide any useful information. Strongly suggest providing the elution volumes.

Responses: As suggested by the reviewer, we have now provided the elution volumes in either figures or legends for all related figures.

x. Fig ED1c: Typo in label: FLAG-BIRC6

Responses: Corrected as suggested, thanks.

xi. Almost all gels and blots lack molecular weight markers (e.g., Fig. ED 1c,d)

Responses: Molecular weight markers are added as suggested, thanks.

xii. Main text and Fig. ED2g "N-terminal session". Do the authors mean "N-terminal section"?

Responses: Corrected as suggested, thanks.

xiii. Fig ED3 i,j: please specify that CBM32 in i) is BIRC6 and that CBM32 is from another protein; which one? Also show reference not just the PDB ID. Same for the PDB ID in k.

Responses: As suggested by the reviewer, protein names have now been provided in the revised Fig. ED3 j and k. We are sorry that the reference to the CBM32 structure CBM BT3015C (PDB: 7BLG) is going to be published.

xiv. Fig ED6c: label myc should be FLAG based on the figure legend.

Responses: Corrected as suggested, thanks.

xv. Page 9, penultimate paragraph: The Fig ED2i should be referenced instead of Fig ED2h.

Responses: Corrected as suggested, thanks.

xvi. Page 15, last line – ‘Extended Data Fig. S6d’ – remove the ‘S’.

Responses: Corrected as suggested, thanks.

xvii. Page 17, line 19 – Cryo-EM is misspelled as cry-EM.

Responses: Corrected as suggested, thanks.

Reviewer 2:

The authors determined the cryo-EM structure of BIRC6, a gigantic 530 kDa protein with little information about its structure known previously. BIRC6 has also been noted to interact with both autophagy and apoptosis pathway. In this manuscript, the authors revealed the cryo-EM structure of BIRC6 at 3.6Å. The authors found that BIRC6 forms a symmetric homodimer with multiple domains in a U-shape structure.

The data on the structural interaction of Smac with BIRC6 was clear. Smac, a previously known ubiquitination substrate of BIRC6, was copurified in the BIRC6 samples, which allowed the authors to elucidate the binding of Smac with BIRC6 in dimeric conformation. Unfortunately, the remaining parts of the manuscript are very unclear and sometimes, digressed far from the first part and with conclusions unjustified or unclear.

Responses: *We agree with the reviewer that the functional part is less well refined. BIRC6 has previously been shown to be important in the regulation of both apoptosis and autophagy (Qiu et al., EMBO J 23: 800-810, 2004; Jia and Bonifacino 2019, Elife 8. 10.7554/eLife.50034; Jiang et al., PNAS, 116: 13404, 2019). In the revised version, we provided additional data and re-structured the manuscript to strengthen our conclusions as advised by the reviewer. To address reviewer's concern on the disconnection between the structural and functional analyses, we have now performed additional experiments to show that (1) Smac regulates the interaction between the active caspase 9 and BIRC6 (revised Fig. 3c) (providing an explanation how structural differences between procaspase 9 and active caspase 9 influence their activities and functions), (2) high concentrations of LIR1 peptides can also outcompete LC3 for binding the LIR motif of BIRC6 (revised Fig. 5d) (providing a link between LIR motif in BIRC6 structure and its function in regulating LC3 as well as autophagy), and (3) the levels of procaspase-9 and BIRC6 are reduced at the same times during autophagy (revised Fig. 4d-e) (strengthening a role of BIRC6 in regulating both apoptosis and autophagy).*

The major issues:

1) The authors also tried to elucidate the structural information of BIRC6 interaction with caspase-9, which was shown previously by the authors. Since the authors have previously reported that BIRC6 can bind with purified processed caspase-9, it is unclear why the authors decided to investigate the binding of unprocessed caspase-9 and BIRC6 in this structural manuscript. It is also unclear why the authors continued to have a section on the interaction of XIAP with procaspase-3, since this is a BIRC6 paper.

Responses: *Unlike other procaspases, procaspase 9 is the initiator caspase, which possesses weak caspase activity to trigger caspase cascade and the ensuing apoptosis. Thus, the regulation of procaspase 9, instead of the processed caspase 9 or other caspases (such as the effector caspase, caspase 3), is critical to the initiation of apoptosis (Cardone et al., 1998, Science 282: 1318-21; Srinivasula et al., Nature 410:112, 2001). As noted correctly by the reviewer, we showed previously that BIRC6*

inhibits the activity of the processed caspase 9, but not processed caspase 3, in vitro (Qiu and Goldberg, J Biol Chem. 280:174, 2005). We also showed previously that BIRC6, unlike XIAP, binds procaspase 9 in vitro in an IBM-independent manner, but was unable to display their association in cells, because BIRC6 can be cleaved by caspase 9 in cells (Qiu and Goldberg, J Biol Chem. 280:174, 2005). By using the uncleavable mutants of procaspase 9 (Procasp 9 3M or 4M), we had finally demonstrated that BIRC6 could co-IP with the transfected procaspase 9 in 293T cell lysates. As shown in this manuscript and three recent reports (Hunkeler et al., Science 379: 1105, 2023; Ehrmann et al., Science 379: 1117, 2023; Dietz et al., Science 379: 1112, 2023), which appeared after our submission, BIRC6 forms a dominant complex with Smac from cells under normal growth conditions. Although other groups reported that BIRC6 binds Smac with much higher affinity than caspase 3 and HtrA2, whether Smac also outcompetes procaspase 9 in cells was unclear. Given that cellular activation of procaspase 9 also requires Apaf-1, dATP and cytochrome c (Roy et al., EMBO J. 16, 6914, 1997), we tested the influence of Smac on the interaction of BIRC6 with procaspase 9 and caspase 3 in the cellular system. Among other IAPs, the BIR3 domain of XIAP has also been shown to bind caspase 9 (Wu et al., Nature 408:1008, 2000; Srinivasula et al., Nature 410:112, 2001). Thus, we employed XIAP as a reference in this study. It turns out that BIRC6 is different from XIAP in the mechanism for associating with Smac (revised Fig. 3b). Thus, we humbly believe that experiments related to caspase 3 and XIAP are important for elucidating the role of the BIRC6-Smac complex in regulating caspase activation/activities and apoptosis, instead of digressing from the structural part of this manuscript.

2) Then the manuscript digressed further into autophagic regulation of caspase-9 levels. It is doubtful that the levels of caspase-9 change in rapamycin treated cells can affect apoptosis (not shown). It is also not clear if the changes in caspase-9 with rapamycin is regulated by BIRC6 (not shown). Thus, it is a big overstatement to say that BIRC6 regulates the balance between autophagy and apoptosis.

Responses: Probably because we did not introduce clearly enough, the reviewer thought that the autophagic regulation of caspase 9 further digressed from the main topic. BIRC6 is so far the only IAP that inhibits both apoptosis and autophagy. Since autophagy antagonizes apoptosis under certain circumstances (Fouillet et al., Autophagy, 8, 915, 2012; Robin et al., Autophagy, 15, 771, 2019), any action of BIRC6 must influence the balance between apoptosis and autophagy. Notably, we found that both procaspase 9 and the processed caspase 9, but not other effector caspases, such as caspase 3, can be degraded by autophagy upon autophagic stimulation, such as rapamycin. These results might provide some clues how the inhibition of autophagy by BIRC6 is removed upon initiation of autophagy. Similarly, BIRC6 can be cleaved by caspases and HtrA2 during apoptosis as we and others demonstrated previously (Qiu and Goldberg, J Biol Chem. 280:174-82, 2005; Sekine et al., Biochem Biophys Res Commun. 330:279, 2005). In order to fill the gap noted by the reviewer, we have now additionally shown that the levels of procaspase-9 and BIRC6 are reduced at the same times by the treatment of rapamycin or starvation in MDA-MB-453 cells in the

revised version (revised Fig. 4d-e).

3) LC3-K51R may have an effect on its expression level or stability, but it is not clear how is that related to BIRC6.

Responses: Probably because we did not explain clearly enough, BIRC6 serves as both a ubiquitin-conjugating enzyme and ubiquitin-ligase (E2/E3) to catalyze monoubiquitylation of LC3 at K51 (Jia and Bonifacino 2019, Elife 8.10.7554/eLife.50034). But, how the BIRC6-mediated ubiquitination of LC3 influences autophagy or apoptosis remained unclear. We had shown that mutation of LC3 at K51 promotes autophagy and autophagic degradation of BIRC6, providing a clue how BIRC6 might regulate autophagy. In the revised version, we have further explained these results (Pages 19-20).

Specifics:

1. The title “Structural basis for BIRC6 to balance apoptosis and autophagy” should be changed. First, it is unclear if apoptosis and autophagy is in any kind of balance or needs to be balanced. Apoptosis promotes cell death; while autophagy promotes degradation and survival. Second, even if there is a balance between apoptosis and autophagy, it is not likely that BIRC6 should be the key regulator as the strongest target of BIRC6 seems to be Smac, which serves a minor role in apoptosis anyway. Third, the authors did not provide data to convincingly demonstrate this point in the manuscript.

Responses: Although we believe that the BIRC6 should be the key regulator for both apoptosis and autophagy, we agree with the reviewer that the data in this manuscript should not be convincing enough to support our points. However, we showed previously that the adaptor protein SIP/CacyBP might balance apoptosis and autophagy probably by controlling the degradation pathways of BIRC6 under certain stress conditions (Jiang et al., PNAS 116: 13404, 2019). Indeed, our data in this manuscript and results by other three groups all demonstrated that BIRC6 forms the strongest complex with Smac among all BIRC6 clients. But, isolations of the stable BIRC6-Smac complex in all these studies were all from cells under normal growth conditions. Either apoptosis or autophagy actually happens under various stress conditions. Procaspase 9 and LC3, both of which are potential targets of BIRC6, are critical to the initiation of apoptosis and autophagy, respectively (Thornberry et al., Science 281, 1312, 1998; Qin et al., Nature, 399:549, 1999; Nakatogawa et al., Cell 130:165, 2007; Xie et al., Mol Biol Cell. 19:3290, 2008) . Thus, we humbly believe that the BIRC6-mediated regulation of procaspase 9 and LC3 or their regulation by Smac should be important to the balance between apoptosis and autophagy. These arguments have also been included in the revised Discussion.

2. The subtitle “LC3 ubiquitination promotes apoptosis” is not justified and should be changed.

Responses: Revised into “K51R mutation of LC3 inhibits apoptosis”, according to reviewer’s comment.

Reviewer 3:

In this study, the authors reveal an excellent cryo-EM structure of mouse BIRC6 at a global resolution of 3.6Å. This structure is organized by two BIRC6 molecules, which arrange in an antiparallel manner to form a U-shaped structure. Based on this structure, authors analyze the organization of different regions/motifs of BIRC6 and the potential interaction fashion of BIRC6 and SMAC. Next, authors demonstrate that procaspase-9 is degraded by autophagy. At the end of this study, authors show that overexpression of LC3 K51R mutant promotes autophagy and inhibits apoptosis. This work provides important insights into how BIRC6 regulates autophagy and apoptosis.

My biggest concern is the disconnect between structural and functional research. Authors illustrate an excellent structure of this giant protein BIRC6, however, this structure is not connected with the function of BIRC6 in regulating autophagy or apoptosis as authors show in the other part of this study. For instance, in page 13, authors claim that LIR1 peptide might change the conformation of BIRC6 LIR, in page 16, authors indicate BIRC6 LIR does not overlap with the binding regions of SMAC. The related structures would be shown to support these hypotheses.

Another major concern is the reproducibility. The quantification of WB and IFA from multiple repeated experiments should be provided.

Responses: *The reviewer kindly praised that we illustrated an excellent structure of BIRC6, and commented that our data on the LC3 K51R mutant-mediated promotion of autophagy and inhibition of apoptosis provided important insights into how BIRC6 regulates autophagy and apoptosis. However, the reviewer concerned the disconnection between structural and functional researches. We principally agree with the reviewer on these comments, and have tried to provide more detailed explanation or additional data to overcome these weaknesses. To address reviewer's concern on the disconnection between the structural and functional analyses, we have now performed additional experiments to show that (1) Smac regulates the interaction between the active caspase 9 and BIRC6 (revised Fig. 3c) (providing an explanation how structural differences between procaspase 9 and active caspase 9 influence their activities and functions), (2) high concentrations of LIR1 peptides can also outcompete LC3 for binding the LIR motif of BIRC6 (revised Fig. 5d) (providing a link between LIR motif in BIRC6 structure and its function in regulating LC3 as well as autophagy), and (3) the levels of procaspase-9 and BIRC6 are reduced at the same times during autophagy (revised Fig. 4d-e) (strengthening a role of BIRC6 in regulating both apoptosis and autophagy). Finally, we repeated certain questionable experiments and provided the quantification of critical immunoblotting experiments as kindly suggested by the reviewer.*

Other specific comments:

1: Figure 2k, BIRC6 is shown to interact with procaspase-9. Does cleaved Cas-9 interact with BIRC6? How does Smac regulate this interaction?

Responses: *We showed previously that BIRC6 also binds the cleaved Casp-9 and inhibits its activity in vitro (Qiu and Goldberg, J Biol Chem. 280:174, 2005). In the*

revised version, we have now shown that unlike procaspase 9, the cleaved caspase 9 can also be outcompeted in the cell lysates by Smac, instead of the Smac mutant (revised Fig. 3c), probably because the cleaved caspase 9 contains an exposed N-terminal IBM.

2. The data of Figure 2a-d show the autophagic degradation of procaspase-9, however, Figure 2e and 2f is about the function of BIRC6 LIR. These are two unrelated contents. Figure 2 should be split to two figures.

Responses: As advised by the reviewer, we have now split Figures into Figures 2a-j and 3a, b and d.

3. Authors conclude that BIRC6 binds LC3 through LIR. As shown in figure 2e,2f. the putative BIRC6 LIR peptide LIR1 is not be able to compete BIRC6 for binding LC3, but a hypothetical universal LIR peptide indeed reduces the interaction of BIRC6 and LC3. This result indicates the putative BIRC6 LIR is not functional. The conclusion need reconsideration.

Responses: We showed previously that this putative BIRC6 LIR could pull-down LC3 in vitro (Jiang et al., PNAS 116: 13404, 2019). Because the hu-LIR peptide sequence is generally different from that of the BIRC6 LIR region and cannot dimerize with the BIRC6 region in the other BIRC6 molecule that dimerizes with BIRC6-LIR, it is thus free to reach the BIRC6 LIR region and block the LC3 binding, which happens at low frequency in regularly cultured cells (Open status). However, BIRC6-LIR1 peptides might competitively bind the part of the other BIRC6 molecule, which supposedly dimerizes with the LIR region of BIRC6 (Closed status), and then make the BIRC6 LIR region free to bind LC3. If this is the case, further increase of LIR1 peptide concentrations would competitively bind the relaxed BIRC6 LIR and reduce the binding of LC3. To validate this hypothesis, we have now used different concentrations of LIR1 peptides in our co-IP experiments and found that LC3 binds BIRC6 at the highest levels in presence of 100 μ M of LIR1 peptides, the binding indeed declines afterwards with the increase of the LIR1 peptide concentrations, and 400 μ M of LIR1 peptide can also outcompete LC3 for binding BIRC6 (revised Fig. 5d). In the revised version, we have now presented a scheme to further explain this model, though it is still speculative (revised Fig. 5e).

4. Figure 4. LC3 null cells need be used to eliminate any effect of endogenous LC3.

Responses: As advised by the reviewer, siRNA of LC3 was used to deplete the endogenous LC3, and results further support our previous conclusion that the LC3 K51R mutation still inhibits the degradation of the transfected GFP-LC3 in the cells with the reduced levels of endogenous LC3 (revised Fig. 6d).

5. Figure 4. Lysine 51 of LC3 is not only be ubiquitinated, but also acetylated. Besides, K51R mutant loses the interaction with autophagy receptor, e.g. p62. Thus, the observed effects by K51R overexpression in this Figure should not be simply attribute to loss of ubiquitination. Authors would exclude other possibilities or the conclusion need

reconsideration

Responses: *We agree with the reviewer on this point and revised our conclusions accordingly (Page 20).*

6. EMD and PDB number should be provided.

Responses: *We have provided the EMD and PDB number as suggested.*

7. A bioRxiv preprint about BIRC6 structure has been cited in this article. Two more similar preprints, which are posted in last August by Moritz Hunkeler and Larissa Dietz, should also be cited and discussed.

Responses: *All these studies have now been published in Science, and we've cited and discussed in the revised version (Page 21) (Hunkeler et al., Science 379: 1105, 2023; Ehrmann et al., Science 379: 1117, 2023; Dietz et al., Science 379: 1112, 2023).*

REVIEWER COMMENTS

Reviewer #1 (Remarks to the Author):

The revised manuscript has been improved in some points, but, unfortunately, some of the key issues highlighted in the 1st round of review remain, and the same of the new additions counteract the presented data. Most importantly these include, i) the disconnect between the BIRC6 structure in the first part of the manuscript and the functional data in the 2nd part, ii) concerns with reproducibility/quality of some of the data, and iii) overstating some of the results, including in the title.

The main points are:

I) The revised manuscript still does not link the BIRC6 structure in the first part of the paper with the functional work on apoptosis and autophagy in the 2nd half. Therefore, the title "Structural basis for BIRC6 to balance apoptosis and autophagy" is misleading and needs to be changed. While an impressive structure due to its size, it does not support any of the functional work described in the paper, highlighted by the fact that no structural details are shown in the 2nd half of the paper.

II) Almost all the data presented in Fig. 3 to support the conclusion that procaspase-9 binding to BIRC6 cannot be outcompeted by Smac and that BIRC6 strongly inhibits caspase-9 but only weakly inhibits caspase-3 is not very convincing or the authors conclusions are difficult to follow. More specifically:

a. l. 217: "Smac with IBM mutations, where the AVPI stretch was mutated into AAAI, further reduced the co-immunoprecipitation of BIRC6 with HtrA2 (Fig. 3b), ...". It is unclear what 'further' refers to in this statement. If the authors aim to compare Smac-WT with Smac-Mut, in this reviewer's view there is no obvious difference in the amount of HtrA2 co-immunoprecipitated with BIRC6 between the Myc-Smac-WT and Myc-Smac-Mut samples. Therefore, the complete newly included section in lines 212 - 221 and the conclusion that "the region other than the IBM of Smac could still outcompete HtrA2 for binding BIRC6." is not supported by data. In general, the differences in some of bands in Fig 3b, specifically HtrA2 and FLAG (Procasp9) are very subtle, especially as the amount of immunoprecipitated BIRC6 varies considerably between different samples, compounding the issues of the minimal differences.

b. l. 229: "Thus, Smac outcompetes caspase-3 and HtrA2, but not procaspase-9 for binding BIRC6 in cells, ...". This statement refers to data presented in Fig. 3b/c. My concern here is that the data in Fig 3b and 3c conflict with each other, suggesting issues with reproducibility. To me, Fig 3b shows that Smac-WT does not affect Procasp9 co-IP (in line with the authors' conclusions), but in Fig 3c, the Procasp9 as well as the Active Casp9 bands seem weaker in the sample containing Myc-Smac-WT compared to Empty vector and Myc-Smac-Mut. As previously commented by Reviewer 3, quantification of multiple WB experiments and careful normalisation would be required to convincingly show these effects.

c. L. 275: The newly inserted section discussing the effects of the hu-LIR peptide is still very difficult to follow. Which part of BIRC6 do the authors refer to by "the corresponding BIRC6 region" (l. 276)? Is this different to the "BIRC6 LIR region" (l. 277)? The authors state that the hu-LIR peptide cannot dimerize with the corresponding BIRC6 region. I did not find any data or reference supporting this statement. If I understand this section correctly, 'bind' may be a better word than 'dimerize' as the binding of a small peptide to a large protein would hardly be considered a dimer.

d. Fig 3d and ED Fig. 4c-e: The authors now present new, markedly differently looking caspase inhibition raw data in ED Fig. 4 c-e compared to the initial manuscript. First, to me it is unclear why the raw data now looks this different compared to the previous data? Have the authors changed their experimental setup? The authors state that BIRC6 strongly inhibits caspase 9, but the inhibition reaches a plateau at less than 20%. This statement should be revised. Further, Dietz et al, Science 2023 show that BIRC6 inhibits caspase 3 activity at low nM concentration. How do the authors explain this discrepancy?

III) l. 91: The authors still describe that the homodimeric BIRC6 structure is "consistent with the elution profile from gel filtration (Extended Data Fig. 1 a-d)." As stated previously, the gel filtration profile presented in Fig. 1c is not sufficient to support this statement as no gel filtration

standard/calibration is shown. Without even this minimum of validation, the statement needs to be removed.

IV) Some of the statements exaggerate the effects observed in the experiments:

a. L. 267: "... the LIR1 peptide at 100 μ M still greatly increase the association of LC3B with BIRC6 in vitro (Fig 5b)." I don't agree with the statement "greatly increases". The effects shown in the Western blot from a gel filtration experiment in Fig. 5b are very subtle and the faint bands barely visible.

V) Fig 5c: Very faint bands as well. It is very hard to appreciate the differences described by the authors as the LC3 bands even in the long exposure are close to the detection limit. As described above for Fig. 3b/c, I am concerned about the reproducibility of these results.

VI) ED Fig 6a,b and text line 306: I do not follow the author's explanation why the PA28 γ band decreases in samples with LC3-K51R but not LC3-WT. If LC3-K51R blocked autophagic degradation of PA28, I would expect a stabilisation of PA28 γ rather than a reduction.

VII) The authors now include the full list of proteins identified by mass spectrometry as a PDF file. This information would be much more useful in a spreadsheet (or .csv file or similar editable file format).

Formatting issues and typos:

- Fig 3a: some of the labels (* and the arrow) seem to point to the wrong space and are shifted upwards.
- Line 268: Typo, BRIC6 should be BIRC6
- Line 275, Fig. 3b should be Fig. 5c.
- ED Fig 3: In the figure legend, JLR1 and JLR2 should be JRL1 and JRL2 respectively.

Reviewer #2 (Remarks to the Author):

I have no more comments.

Reviewer #3 (Remarks to the Author):

The authors did a commendable job in addressing the points raised in the first round of review. However their responses to some comments are still problematic.

1. Regarding to my concern about the connection between structural and the functional study, this is still not clarified in the revision. Authors claimed that they have explained the structural differences between procaspase 9 and active caspase 9. However, no structural information of pro/active caspase-9 was shown.

2. Authors tested different doses of LIR1 peptides, but there is still no structure evidence to support their hypothesis shown in Fig 5e. (1) The position of LIR should be shown in the structure model of BIRC6. (2) LIR1 peptide is shown to bind LIR of BIRC6. Is this supported by experimental evidence? (3)The structure data indicates BIRC6 forms an anti-parallel dimer, however, in fig 5e, two BIRC6 molecules presents as parallel dimer. This LIR binding model is questionable.

3. Authors conducted LC3 knockdown by siRNA transfection, as shown in fig 6d, the knockdown efficiency is questionable. Authors would try CRISPR.

4. In figure 6a, b, c, e, f and g, LC3-WT, LC3-K51R overexpression should be performed in LC3-null background to eliminate any effect of endogenous LC3.

5. Line 309, "LC3-K51R also blocked autophagic degradation of PA28 γ ", Line 314 "this mutation (K51R) promoted autophagic degradation of BIRC6 and the autophagy receptor protein p62". What's the exactly role of K51R in regulating autophagic degradation?

6. Figure 6e, the antibody for endogenous-LC3 is supposed to stain the transfected GFP-LC3 as well. The fluorescence in the endogenous-LC3 panel seems come from the staining of GFP-LC3. How did authors distinct endogenous LC3 and GFP-LC3?

Re: Manuscript NCOMMS-22-53213A, “Structural basis for BIRC6 to balance apoptosis and autophagy” by Liu et al.

Point-to-point responses to reviewers

Reviewer 1:

The revised manuscript has been improved in some points, but, unfortunately, some of the key issues highlighted in the 1st round of review remain, and the same of the new additions counteract the presented data. Most importantly these include, i) the disconnect between the BIRC6 structure in the first part of the manuscript and the functional data in the 2nd part, ii) concerns with reproducibility/quality of some of the data, and iii) overstating some of the results, including in the title.

Responses: Thank the reviewer for these important comments. i) The reviewer’s concern with the disconnection between two parts of the manuscript has been addressed by changing the title as detailed below. ii) We have performed more repeated experiments with the revised settings, and have improved the quality of Fig. 3b and 5c with reproducible results. In addition, we have added a new experiment to show that the LIR1 peptide of BIRC6 can bind the putative LIR-pairing region of the anti-parallel dimer of BIRC6 molecules (ED Fig. 5e). iii) With these changes, we humbly believe that there is no more overstatement in the results or the title.

The main points are:

I) The revised manuscript still does not link the BIRC6 structure in the first part of the paper with the functional work on apoptosis and autophagy in the 2nd half. Therefore, the title “Structural basis for BIRC6 to balance apoptosis and autophagy” is misleading and needs to be changed. While an impressive structure due to its size, it does not support any of the functional work described in the paper, highlighted by the fact that no structural details are shown in the 2nd half of the paper.

Responses: The original title, which just states structural basis of BIRC6, actually makes any functional data disconnected. Thanks to the reviewer’s advice, we have now changed the title into “Molecular mechanisms underlying the BIRC6-mediated regulation of apoptosis and autophagy” with a hope to cover both structural and functional studies.

II) Almost all the data presented in Fig. 3 to support the conclusion that procaspase-9 binding to BIRC6 cannot be outcompeted by Smac and that BIRC6 strongly inhibits caspase-9 but only weakly inhibits caspase-3 is not very convincing or the authors

conclusions are difficult to follow. More specifically:

a. l. 217: “Smac with IBM mutations, where the AVPI stretch was mutated into AAAI, further reduced the co-immunoprecipitation of BIRC6 with HtrA2 (Fig. 3b), ...”. It is unclear what ‘further’ refers to in this statement. If the authors aim to compare Smac-WT with Smac-Mut, in this reviewer’s view there is no obvious difference in the amount of HtrA2 co-immunoprecipitated with BIRC6 between the Myc-Smac-WT and Myc-Smac-Mut samples. Therefore, the complete newly included section in lines 212 – 221 and the conclusion that “the region other than the IBM of Smac could still outcompete HtrA2 for binding BIRC6.” is not supported by data. In general, the differences in some of bands in Fig 3b, specifically HtrA2 and FLAG (Procas9) are very subtle, especially as the amount of immunoprecipitated BIRC6 varies considerably between different samples, compounding the issues of the minimal differences.

Responses: Thanks for the reviewer’s critical comments, we have now repeated the experiments by adding more lysates in the IP assay, and found that Smac with IBM mutations, where the AVPI stretch was mutated into AAAI, could not reduce the co-immunoprecipitation of BIRC6 with HtrA2 (the revised Fig. 3b, e), suggesting that the IBM of HtrA2 should also mediate its association with BIRC6. These results are reproducible, and a quantification of multiple WB experiments is included in the revised Fig. 3b-e.

The word “further” was mistakenly used here, and has been dropped now.

b. l. 229: “Thus, Smac outcompetes caspase-3 and HtrA2, but not procaspase-9 for binding BIRC6 in cells, ...”. This statement refers to data presented in Fig. 3b/c. My concern here is that the data in Fig 3b and 3c conflict with each other, suggesting issues with reproducibility. To me, Fig 3b shows that Smac-WT does not affect Procasp9 co-IP (in line with the authors’ conclusions), but in Fig 3c, the Procasp9 as well as the Active Casp9 bands seem weaker in the sample containing Myc-Smac-WT compared to Empty vector and Myc-Smac-Mut. As previously commented by Reviewer 3, quantification of multiple WB experiments and careful normalisation would be required to convincingly show these effects.

Responses: We humbly believe that the data in Fig. 3b and Fig. 3c (the revised Fig. 3f) did not conflict with each other, because these are two totally different experimental systems. In Fig. 3b, procaspase 9 was co-IPed in the regular cell lysates. In contrast, in order to obtain active caspase 9 in the cell lysates, the caspase cascade was activated by adding Cyt c and dATP into the lysates of the cells without any transfection in Fig. 3c (the revised Fig. 3f). Then, the lysates with the activated caspases were mixed with the lysates of cells that were transfected with Myc-Smac. Transfection of proapoptotic Smac reduced levels of procaspase 9, but elevated levels of active caspase 9 in the input as stated in the previously revised legend of Fig. 3c (the revised Fig. 3f). Eventually, Smac, but not its mutant, still reduced the levels of active caspase 9 co-IPed with BIRC6. As suggested by the reviewer, quantification of Fig.3c has now been included in the newly revised version.

c. L. 275: The newly inserted section discussion the effects of the hu-LIR peptide is still very difficult to follow. Which part of BIRC6 do the authors refer to by “the corresponding BIRC6 region” (l. 276)? Is this different to the “BIRC6 LIR region” (l. 277)? The authors state that the hu-LIR peptide cannot dimerize with the corresponding BIRC6 region. I did not find any data or reference supporting this statement. If I understand this section correctly, ‘bind’ may be a better word than ‘dimerize’ as the binding of a small peptide to a large protein would hardly be considered a dimer.

Responses: *As described at lines 259-260, BIRC6 contains a putative LIR (amino acids 4670-NPQ TSS FLQV LV-4681 in mouse). Because BIRC6 forms an anti-parallel dimer, the sequence of the corresponding region of the other molecule of BIRC6, which binds the LIR sequence in the dimer and is now named as the LIR-pairing region, is different from the LIR region. Thus, we speculate that the LIR1 peptide should preferentially pair with the LIR-pairing region in the other molecule of BIRC6. On the other hand, hu-LIR peptide with a sequence different from any known protein (hu-LIR, EPLDFDWEIVLEEEM), which theoretically competes with any LIR motif (ref. 40 in this manuscript), should not pair with the LIR-pairing region of BIRC6, because the hu-LIR peptide sequence is generally different from that of the BIRC6 LIR region. So, the hu-LIR peptide can competitively block LC3 to bind LIR. However, LIR1 at low concentrations might prefer to pair with the LIR-pairing region to relax the LIR region from the dimer, resulting in the increased binding of LC3 to BIRC6. Notably, excessive LIR1 peptides (i.e., high concentrations) should also competitively block the LC3 binding to the LIR region. In the newly revised version, we have added a new experiment to show that the LIR1 peptide of BIRC6 can bind the putative LIR-pairing region of the anti-paralleled dimer of BIRC6 molecules (ED Fig. 5e), and have tried to define these players in more details to avoid any possible confusion. Ref. 40 should explain why the hu-LIR peptide cannot dimerize with the LIR-pairing region, because the hu-LIR peptide sequence is different from that of any known protein.*

d. Fig 3d and ED Fig. 4c-e: The authors now present new, markedly differently looking caspase inhibition raw data in ED Fig. 4 c-e compared to the initial manuscript. First, to me it is unclear why the raw data now looks this different compared to the previous data? Have the authors changed their experimental setup? The authors state that BIRC6 strongly inhibits caspase 9, but the inhibition reaches a plateau at less than 20%. This statement should be revised. Further, Dietz et al, Science 2023 show that BIRC6 inhibits caspase 3 activity at low nM concentration. How do the authors explain this discrepancy?

Responses: *In the first version of the manuscript, we measured initial rates of caspase activities with 4 different concentrations of BIRC6 within 1800 seconds, whereas we recorded reactions with 9 different concentrations of BIRC6 for 4200 seconds until the reactions reached a plateau in the previously revised manuscript. Although they look so differently, the conclusions are similar because only the initial rates can reflect the reaction rates. So, even though the inhibition reaches the plateau at less than 20%, the difference has been determined before the plateau is reached.*

We note that Dietz et al (Science 2023) used purified enzymes to show inhibition of caspase 3 activity by BIRC6, a system totally different from the cell lysates in our studies. Therefore, our results don't contradict with theirs. Additionally, we compared the difference between caspase 9 and caspase 3 in response to inhibition of BIRC6 in a physiology-relevant cell lysate system.

III) 1. 91: The authors still describe that the homodimeric BIRC6 structure is “consistent with the elution profile from gel filtration (Extended Data Fig. 1 a-d).” As stated previously, the gel filtration profile presented in Fig. 1c is not sufficient to support this statement as no gel filtration standard/calibration is shown. Without even this minimum of validation, the statement needs to be removed.

Responses: As suggested by the reviewer, this statement has now been removed. As explained last time, the presented SEC profile in ED Fig. 1b did not show whether BIRC6 is a dimer, because we did not expect it until observing it by Cryo-EM. Sorry for failing to change it last time.

IV) Some of the statements exaggerate the effects observed in the experiments:

a. L. 267: “... the LIR1 peptide at 100 μ M still greatly increase the association of LC3B with BIRC6 in vitro (Fig 5b).” I don't agree with the statement “greatly increases”. The effects shown in the Western blot from a gel filtration experiment in Fig. 5b are very subtle and the faint bands barely visible.

Responses: We have dropped the word “greatly”, though we consider the increase very obvious because the increase was diluted in many collection tubes.

V) Fig 5c: Very faint bands as well. It is very hard to appreciate the differences described by the authors as the LC3 bands even in the long exposure are close to the detection limit. As described above for Fig. 3b/c, I am concerned about the reproducibility of these results.

Responses: We have performed more repeated experiments with the revised settings and have improved the quality of Fig. 5c with reproducible results. A quantification of multiple WB experiments is included in the revised version.

VI) ED Fig 6a,b and text line 306: I do not follow the author's explanation why the PA28 γ band decreases in samples with LC3-K51R but not LC3-WT. If LC3-K51R blocked autophagic degradation of PA28, I would expect a stabilisation of PA28 γ rather than a reduction.

Responses: Sorry, it was a mistake. LC3 K51R should have “promoted”, instead of “blocked”, autophagic degradation of PA28 γ .

VII) The authors now include the full list of proteins identified by mass spectrometry as a PDF file. This information would be much more useful in a spreadsheet (or .csv file or similar editable file format).

Responses: Raw mass spectrometry data in .csv file and a spreadsheet in Excel format

had been uploaded on ProteomeXchange Datasets with the accession number PXD042246, when the previously revised version was submitted. We are sorry for failing to provide the related information in previously revised version, but we provide it now.

Formatting issues and typos:

- Fig 3a: some of the labels (* and the arrow) seem to point to the wrong space and are shifted upwards.

Responses: Corrected as suggested, thanks.

- Line 268: Typo, BRIC6 should be BIRC6

Responses: Corrected as suggested, thanks.

- Line 275, Fig. 3b should be Fig. 5c.

Responses: Corrected as suggested, thanks.

- ED Fig 3: In the figure legend, JLR1 and JLR2 should be JRL1 and JRL2 respectively.

Responses: Corrected as suggested, thanks.

Reviewer #2:

I have no more comments.

Reviewer #3:

The authors did a commendable job in addressing the points raised in the first round of review. However their responses to some comments are still problematic.

1. Regarding to my concern about the connection between structural and the functional study, this is still not clarified in the revision. Authors claimed that they have explained the structural differences between procaspase 9 and active caspase 9. However, no structural information of pro/active caspase-9 was shown.

Responses: *In our previous responses to the reviewer, the structural differences between procaspase 9 and active caspase were speculated based on their differences in primary sequences and functions, though we were unable to show. Thus, we did not claim it in the revised manuscript. The original title, which just states structural basis of BIRC6, actually makes any functional data disconnected. We have now changed the title into “Molecular mechanisms underlying the BIRC6-mediated regulation of apoptosis and autophagy” with a hope to cover both structural and functional studies.*

2. Authors tested different doses of LIR1 peptides, but there is still no structure evidence to support their hypothesis shown in Fig 5e. (1) The position of LIR should be shown in the structure model of BIRC6. (2) LIR1 peptide is shown to bind LIR of BIRC6. Is this supported by experimental evidence? (3) The structure data indicates BIRC6 forms an anti-parallel dimer, however, in fig 5e, two BIRC6 molecules presents as parallel dimer. This LIR binding model is questionable.

Responses: *(1) As suggested by the reviewer, LIR (amino acids 4670-NPQ TSS FLQV LV-4681) position is now labelled in the structure model of BIRC6 at ED Fig. 6.*

(2) We showed previously that GST-LIR could associate with LC3 (Jiang et al., PNAS, 2019), but we didn't test whether LIR1 peptide really binds the putative LIR-pairing region. In the revised version, we have added a new experiment to show that the LIR1 peptide of BIRC6 can bind the putative LIR-pairing region of the anti-paralleled dimer of BIRC6 molecules (ED Fig. 5e).

(3) In Fig. 5e, we attempted to present an anti-parallel dimer, but didn't make it clear. In the newly revised version, we have improved presentation and labeled N- and C-ends of each molecule.

3. Authors conducted LC3 knockdown by siRNA transfection, as shown in fig 6d, the knockdown efficiency is questionable. Authors would try CRIPSR.

Responses: *The reviewer correctly noted the limited efficiency of LC3 knockdown in Fig. 6d (ED Fig. 6d in the newly revised version) and suggested us to try CRISPR. We have now added the data to show that LC3-K51R plays the similar role in the LC3-deficient human lung cancer A549 cell line obtained by CRISPR, a kind gift from Prof. Qinfang Liu at Chinese Academy of Agricultural Sciences (revised Fig. 6).*

4. In figure 6a, b, c, e, f and g, LC3-WT, LC3-K51R overexpression should be performed in LC3-null background to eliminate any effect of endogenous LC3.

Responses: *As explained above, we have now shown that LC3-K51R plays the similar role in the LC3-deficient cells in the revised Fig. 6g, h, suggesting that the results in Fig. 6a, b, c, e, f and g are all due to the mutation of LC3-K51R.*

5. Line 309, "LC3-K51R also blocked autophagic degradation of PA28 γ ", Line 314 "this mutation (K51R) promoted autophagic degradation of BIRC6 and the autophagy receptor protein p62". What's the exactly role of K51R in regulating autophagic degradation?

Responses: *Sorry, it was a mistake. LC3 K51R should have "promoted", instead of "blocked", autophagic degradation of PA28 γ .*

6. Figure 6e, the antibody for endogenous-LC3 is supposed to stain the transfected GFP-LC3 as well. The fluorescence in the endogenous-LC3 panel seems come from the staining of GFP-LC3. How did authors distinct endogenous LC3 and GFP-LC3?

Responses: *The LC3 antibody can visualize the endogenous LC3 in addition to GFP-LC3, whereas the GFP-LC3 panel cannot show the endogenous LC3. By comparing the difference between the LC3 antibody and GFP-LC3 panels, we can theoretically determine which autophagosomes (or LC3 puncta) are from endogenous LC3 or from the transfected GFP-LC3. Upon treatment of rapamycin, most puncta are yellow after merging two panels, suggesting that expression of the transfected GFP-LC3 is dominant. However, when GFP-LC3 K51R was transfected, many puncta are still green after merging, suggesting that the antibody staining is not efficient enough. An additional note has been added in the newly revised legend of Fig. 6i.*

REVIEWER COMMENTS

Reviewer #1 (Remarks to the Author):

The authors have now further improved the manuscript and have addressed most of my concerns with better-quality experiments and quantification of key data, therefore now alleviating my previous concerns on strength and reproducibility of most of the data.

The paper still has some weaknesses, in particular the section "BIRC6 binds LC3 through LIR" which is still difficult to follow and the presented data not fully convincing.

Nevertheless, as a whole, I think the paper reports noteworthy results and the data support the main conclusion.

As a final point, the authors should check the XIAP data Fig. 3d. I find it surprising that there is a significant difference between empty vector (1st bar) and Flag-procasp9-4M+SMAC-WT (3rd bar) but not between empty vector and the other 2 conditions, even though the latter have the same or much higher average relative levels. In addition, the relevant text (ll. 209–211) states "Overexpression of the Myc-tagged wild-type Smac did not affect the association of BIRC6 or XIAP with the FLAG-tagged quadruple mutants of procaspase 9, but markedly reduced their association with the pro-apoptotic protease HtrA2/Omi or procaspase 3 (Fig. 3b-e), ...". This statement is correct for BIRC6, but the presented data in Fig. 3d does not show that association between HtrA2 and XIAP is reduced by WT Smac. This statement needs to be corrected.

Reviewer #3 (Remarks to the Author):

Thanks to the authors for responding to my comments. The quality of the manuscript has been improved. My biggest concern remains the disconnect between structural and functional studies, which was not fully addressed in the last revision.

1. Figure 3. Authors shown that BIRC6 interacts with Procasp9-4M, and analyzed the regulation of SMAC on this interaction through immunoprecipitation assay. Could author analyze the interaction pattern of these three proteins from structural perspective? e.g., showing the interaction interfaces of the above interactions and analyzing their regulation. This would be a good example to show the connection between structural data and functional analysis, which is the weakness of this article.

2. Figure 6e. MTORC1 inhibition (autophagy activation) by rapamycin reduced the levels of GFP-LC3-I for both WT and K51R, probably due to the conversion of GFP-LC3-I to LC3-II). In figure 6f, the similar treatment with rapamycin seemed not change the levels of GFP-LC3 and GFP-K51R.

3. Figure 5. Authors "hypothesize that BIRC6-LIR1 peptides competitively and preferentially bind the part of the other BIRC6 molecule (i.e., the LIR-pairing region)". Pulldown results shown that N1 region aa (55-386) interact with LIR1 peptide in vitro. As mentioned in comment 1, Could authors provide structural evidence to confirm the interaction between LIR1 and N1 region? Can author see the interaction interface of these two regions from the structure of BIRC6 dimer? Such structure analysis would improve the manuscript and build connection between the structure and the function of BIRC6.

Re: Manuscript NCOMMS-22-53213C-Z, “Molecular mechanisms underlying the BIRC6-mediated regulation of apoptosis and autophagy” by Liu et al.

Point-to-point responses to reviewers

Reviewer #1:

The authors have now further improved the manuscript and have addressed most of my concerns with better-quality experiments and quantification of key data, therefore now alleviating my previous concerns on strength and reproducibility of most of the data.

The paper still has some weaknesses, in particular the section “BIRC6 binds LC3 through LIR” which is still difficult to follow and the presented data not fully convincing.

Nevertheless, as a whole, I think the paper reports noteworthy results and the data support the main conclusion.

As a final point, the authors should check the XIAP data Fig. 3d. I find it surprising that there is a significant difference between empty vector (1st bar) and Flag-procasp9-4M+SMAC-WT (3rd bar) but not between empty vector and the other 2 conditions, even though the latter have the same or much higher average relative levels. In addition, the relevant text (ll. 209–211) states “Overexpression of the Myc-tagged wild-type Smac did not affect the association of BIRC6 or XIAP with the FLAG-tagged quadruple mutants of procaspase 9, but markedly reduced their association with the pro-apoptotic protease HtrA2/Omi or procaspase 3 (Fig. 3b-e), ...”. This statement is correct for BIRC6, but the presented data in Fig. 3d does not show that association between HtrA2 and XIAP is reduced by WT Smac. This statement needs to be corrected.

Responses: Thanks to the reviewer for this important issue. Our results in Fig. 3d were calculated from newly repeated experiments, which are different from our previous judgement. Indeed, the statement for XIAP here needs to be corrected. In the revised version, we have corrected the statement by dropping the word “XIAP”.

Reviewer #3:

Thanks to the authors for responding to my comments. The quality of the manuscript has been improved. My biggest concern remains the disconnect between structural and functional studies, which was not fully addressed in the last revision.

Responses: Thanks to the reviewer for the additional comments. Unlike the original

title that just states structural basis of BIRC6, we humbly believe that the revised title “Molecular mechanisms underlying the BIRC6-mediated regulation of apoptosis and autophagy” should allow us to include both structural and functional data of the related studies. But, we still attempted to analyze the additional structural interactions as advised by the reviewer.

1. Figure 3. Authors shown that BIRC6 interacts with Procasp9-4M, and analyzed the regulation of SMAC on this interaction through immunoprecipitation assay. Could author analyze the interaction pattern of these three proteins from structural perspective? e.g., showing the interaction interfaces of the above interactions and analyzing their regulation. This would be a good example to show the connection between structural data and functional analysis, which is the weakness of this article.

Responses: Thanks to the reviewer for the advice. As advised, we attempted to predict the interaction pattern between BIRC6 and Procasp9 using AlphaFold2-Multimer. Since the sum of the residue sequences of BIRC6 dimer and Procasp9 dimer is too long, the global structure of BIRC6-Procasp9 could not be directly predicted due to limitations of computational resources. Instead, we predicted the structures of the complex of Procasp9 with special BIRC6 domains. According to our structural observations and previous studies, two domains of BIRC6 contribute to the binding to SMAC: BIR domain (interacting with the N-terminal IBM of SMAC) and CBM32 domain. SMAC probably makes use of these two interactions to outcompete caspase 3 and HtrA2. Whether SMAC could regulate the binding of Procasp9-4M with BIRC6 in a same way is unknown. Given that the IBM of Procasp9 is only exposed after proteolytic activation, the IBM-BIR interaction should not be established in the form Procasp9. Therefore, we predicted the structures of the complex of Procasp9 with CBM32 of BIRC6. But no credible interface was identified, partially indicating that Procasp9 might interact with BIRC6 in a different way from that of SMAC or mediated by other factors. This structural prediction is consistent with the conclusion deduced from the biochemical assays “Smac cannot reduce procaspase 9 binding”.

2. Figure 6e. MTORC1 inhibition (autophagy activation) by rapamycin reduced the levels of GFP-LC3-I for both WT and K51R, probably due to the conversion of GFP-LC3-I to LC3-II). In figure 6f, the similar treatment with rapamycin seemed not change the levels of GFP-LC3 and GFP-K51R.

Responses: Thanks to the reviewer for this careful examination and comment. In Figure 6e, Bafilomycin A1 (Baf A1) was added to visualize the conversion of LC3-I to LC3-II, because Baf A1 is an inhibitor of lysosomal and endosomal acidification that blocks the fusion of autophagosomes with endosomes or lysosomes (Rubinsztein et al., Autophagy 5, 585–9, 2009). As expected, treatment with Baf A1 resulted in detectable LC3-II in Figure 6e, whereas LC3-II was not detectable in Figure 6f, where Baf A1 was not added. To improve our presentation, an explanation has been added in the revised text.

3. Figure 5. Authors "hypothesize that BIRC6-LIR1 peptides competitively and preferentially bind the part of the other BIRC6 molecule (i.e., the LIR-pairing region)". Pulldown results shown that N1 region aa (55-386) interact with LIR1 peptide in vitro. As mentioned in comment 1, Could authors provide structural evidence to confirm the interaction between LIR1 and N1 region? Can author see the interaction interface of these two regions from the structure of BIRC6 dimer? Such structure analysis would improve the manuscript and build connection between the structure and the function of BIRC6.

Responses: We thank this reviewer for this constructive suggestion. As advised, we have carefully checked the local density map of the N-terminal section. However, due to the relatively lower local resolution of this section (6 Å, as shown in Extended Data Fig. 2h) and the small molecular size of LIR1, we were unable to unambiguously identify the density of the associated LIR1. To address this issue, we predicted the structure of the complex of the LIR1 peptide and the N-terminal section of BIRC6 using AlphaFold2-Multimer (see Methods). The predicted models revealed an interaction between the LIR1 peptide and the N1 region of BIRC6, characterized by the formation an antiparallel β -sheet interface involving LIR1 and residues 336-339 of N1. This predicted interaction is highly consistent with the results of the pull-down assays, and have been depicted in the revised Extended Data Fig. 5f and in the revised text.